# P²OT: Progressive Partial Optimal Transport for Deep Imbalanced Clustering

**Chuyu Zhang**[1,2,*]    **Hui Ren**[1,*]  **Xuming He**[1,3]
[1]ShanghaiTech University, Shanghai, China    [2]Lingang Laboratory, Shanghai, China
[3]Shanghai Engineering Research Center of Intelligent Vision and Imaging, Shanghai, China
`{zhangchy2,renhui,hexm}@shanghaitech.edu.cn`

## Abstract

Deep clustering, which learns representation and semantic clustering without labels information, poses a great challenge for deep learning-based approaches. Despite significant progress in recent years, most existing methods focus on uniformly distributed datasets, significantly limiting the practical applicability of their methods. In this paper, we first introduce a more practical problem setting named deep imbalanced clustering, where the underlying classes exhibit an imbalance distribution. To tackle this problem, we propose a novel pseudo-labeling-based learning framework. Our framework formulates pseudo-label generation as a progressive partial optimal transport problem, which progressively transports each sample to imbalanced clusters under prior distribution constraints, thus generating imbalance-aware pseudo-labels and learning from high-confident samples. In addition, we transform the initial formulation into an unbalanced optimal transport problem with augmented constraints, which can be solved efficiently by a fast matrix scaling algorithm. Experiments on various datasets, including a human-curated long-tailed CIFAR100, challenging ImageNet-R, and large-scale subsets of fine-grained iNaturalist2018 datasets, demonstrate the superiority of our method.

## 1 Introduction

Human beings possess the innate ability to categorize similar concepts, even when encountering them for the first time. In contrast, artificial models struggle to group similar concepts without labels, although they excel primarily in scenarios where extensive semantic labeling is available. Deep clustering, which aims to learn representation and semantic clustering, is proposed to enable models to achieve that. Currently, many works (Ji et al., 2019; Van Gansbeke et al., 2020; Ronen et al., 2022) are advancing the field of deep clustering. However, they focus on developing methods on balanced datasets, limiting their application in reality, where the data distributions are imbalanced.

In this paper, we propose a practical deep imbalanced clustering problem, bridging the gap between the existing deep clustering and real-world application. Existing methods are broadly categorized into relation matching-based (Dang et al., 2021), mutual information maximization-based (Ji et al., 2019), and pseudo-labeling (PL)-based (Van Gansbeke et al., 2020). They face significant challenges when dealing with imbalanced clustering. Relation matching and mutual information maximization act as surrogate losses for clustering and show inferior results compared to the PL method (Van Gansbeke et al., 2020; Niu et al., 2022). PL, on the other hand, tends to learn degenerate solutions (Caron et al., 2018) and heavily relies on the quality of the pseudo-labels. This reliance often requires an additional training stage for representation initialization and sensitive hyperparameter tuning to mitigate confirmation bias (Arazo et al., 2020; Wei et al., 2021), especially in challenging imbalanced clustering scenarios, where imbalanced learning will seriously affect the quality of pseudo labels.

To mitigate the above weakness, we propose a novel PL-based progressive learning framework for deep imbalanced clustering. Our framework introduces a novel Progressive Partial Optimal Transport (P²OT) algorithm, which integrates imbalance class distribution modeling and confident sample selection within a single optimization problem, enabling us to generate imbalanced pseudo labels and learn from high-confident samples in a progressive manner. Computationally, we reformulate our P²OT as a typical unbalanced OT problem with a theoretical guarantee and employ the light-speed scaling algorithm (Cuturi, 2013; Chizat et al., 2018) to solve it.

---

*Both authors contributed equally. Code is available at https://github.com/rhfeiyang/PPOT.

Specifically, in our progressive PL-based learning framework, we leverage cluster estimation to generate soft pseudo-labels while imposing a set of prior constraints, including an inequality constraint governing sample weight, a Kullback-Leibler ($KL$) divergence constraint on cluster size uniform distribution, and a total mass constraint. Notably, the $KL$ divergence-based uniform distribution constraint empowers our method to avoid degenerate solutions. And our approach with $KL$, which represents a relaxed constraint compared to an equality constraint, facilitates the generation of imbalanced pseudo-labels. The total mass constraint allows us to selectively learn from high-confident samples by adjusting their weights through optimization, alleviating the noisy pseudo-label learning, and eliminating the sensitive and hand-crafted confidence threshold tuning. As training progresses, we incrementally raise the total mass value, facilitating a gradual transition from learning easy samples to tackling more difficult ones. To solve our P$^2$OT problem efficiently, we introduce a virtual cluster and replace the naive $KL$ with the weighted $KL$ constraint, thus reformulating our P$^2$OT into an unbalanced optimal transport problem under transformed constraints. Then, we solve the unbalanced OT with a scaling algorithm that relies on light-speed matrix-vector multiplications.

To validate our method, we propose a new challenging benchmark, which encompasses a diverse range of datasets, including the human-curated CIFAR100 dataset (Krizhevsky et al., 2009), challenging 'out-of-distribution' ImageNet-R dataset (Hendrycks et al., 2021), and large-scale subsets of fine-grained iNaturalist18 dataset (Van Horn et al., 2018). Experiments on those challenging datasets demonstrate the superiority of our method. In summary, our contribution is as follows:

- We generalize the deep clustering problem to more realistic and challenging imbalance scenarios, and establish a new benchmark.

- We propose a novel progressive PL-based learning framework for deep imbalance clustering, which formulates the pseudo label generation as a novel P$^2$OT problem, enabling us to consider class imbalance distribution and progressive learning concurrently.

- We reformulate the P$^2$OT problem as an unbalanced OT problem with a theoretical guarantee, and solve it with the efficient scaling algorithm.

- Our method achieves the SOTA performance on most datasets compared with existing methods on our newly proposed challenging and large-scale benchmark.

## 2 RELATED WORK

**Deep clustering.** The goal of deep clustering (Zhou et al., 2022a; Huang et al., 2022) is to learn representation and cluster data into semantic classes simultaneously. Based on clustering methods, current works can be grouped into three categories: relation matching, mutual information maximization, and pseudo labeling. Specifically, relation matching (Chang et al., 2017; Van Gansbeke et al., 2020; Tao et al., 2020) involves the minimization of the distance between instances considered 'similar' by representation distance, thus achieving clustering in a bottom-up way. Mutual information maximization (Ji et al., 2019; Li et al., 2021; Shen et al., 2021) either in the prediction or representation space, aims at learning invariant representations, thereby achieving clustering implicitly (Ben-Shaul et al., 2023). Pseudo labeling (PL) (Lee et al., 2013), which is widely used in semi-supervised learning (SSL) (Sohn et al., 2020; Zhang et al., 2021a), assigns each instance to semantic clusters and has demonstrated exceptional performance in the domain of deep clustering (Caron et al., 2018; Van Gansbeke et al., 2020; Asano et al., 2020; Niu et al., 2022). In contrast to the relatively clean pseudo-labels produced by SSL, deep clustering often generates noisier pseudo-labels, particularly in scenarios characterized by data imbalance, imposing a great challenge on pseudo-label generation. In this paper, our primary focus is on PL, and we introduce a novel P$^2$OT algorithm. This algorithm progressively generates high-quality pseudo labels by simultaneously considering class imbalance distribution and sample confidence through a single convex optimization problem.

**Optimal transport and its application.** Optimal Transport (OT) (Villani et al., 2009; Peyré et al., 2017; Chizat et al., 2018; Khamis et al., 2023) aims to find the most efficient transportation plan while adhering to marginal distribution constraints. It has been used in a broad spectrum of machine learning tasks, including but not limited to generative model (Frogner et al., 2015; Gulrajani et al., 2017), semi-supervised learning (Tai et al., 2021; Taherkhani et al., 2020), clustering (Asano et al., 2020; Caron et al., 2020), and domain adaptation (Flamary et al., 2016; Chang et al., 2022; Liu et al., 2023; Chang et al., 2023). Of particular relevance to our work is its utilization in pseudo labeling (Asano et al., 2020; Tai et al., 2021; Zhang et al., 2023). Asano et al. (2020) initially

introduces an equality constraint on cluster size, formulating pseudo-label generation as an optimal transport problem. Subsequently, Tai et al. (2021) enhances the flexibility of optimal transport by incorporating relaxation on the constraint, which results in its failure in deep imbalanced clustering, and introduces label annealing strategies through an additional total mass constraint. Recently, Zhang et al. (2023) relaxes the equality constraint to a $KL$ divergence constraint on cluster size, thereby addressing imbalanced data scenarios. In contrast to these approaches, which either ignore class imbalance distribution or the confidence of samples, our novel P$^2$OT algorithm takes both into account simultaneously, allowing us to generate pseudo-labels progressively and with an awareness of class imbalance. In terms of computational efficiency, we transform our P$^2$OT into an unbalanced optimal transport (OT) problem under specific constraints and utilize a light-speed scaling algorithm.

## 3  PRELIMINARY

Optimal Transport (OT) is the general problem of moving one distribution of mass to another with minimal cost. Mathematically, given two probability vectors $\boldsymbol{\mu} \in \mathbb{R}^{m \times 1}, \boldsymbol{\nu} \in \mathbb{R}^{n \times 1}$, as well as a cost matrix $\mathbf{C} \in \mathbb{R}^{m \times n}$ defined on joint space, the objective function which OT minimizes is as follows:

$$\min_{\mathbf{Q} \in \mathbb{R}^{m \times n}} \langle \mathbf{Q}, \mathbf{C} \rangle_F + F_1(\mathbf{Q}\mathbf{1}_n, \boldsymbol{\mu}) + F_2(\mathbf{Q}^\top \mathbf{1}_m, \boldsymbol{\nu}) \tag{1}$$

where $\mathbf{Q} \in \mathbb{R}^{m \times n}$ is the transportation plan, $\langle, \rangle_F$ denotes the Frobenius product, $F_1, F_2$ are constraints on the marginal distribution of $\mathbf{Q}$ and $\mathbf{1}_n \in \mathbb{R}^{n \times 1}, \mathbf{1}_m \in \mathbb{R}^{m \times 1}$ are all ones vector. For example, if $F_1, F_2$ are equality constraints, i.e. $\mathbf{Q}\mathbf{1}_n = \boldsymbol{\mu}, \mathbf{Q}^\top \mathbf{1}_m = \boldsymbol{\nu}$, the above OT becomes a widely-known Kantorovich's form (Kantorovich, 1942). And if $F_1, F_2$ are $KL$ divergence or inequality constraints, Equ.(1) turns into the unbalanced OT problem (Liero et al., 2018).

In practice, to efficiently solve Kantorovich's form OT problem, Cuturi (2013) introduces an entropy term $-\epsilon \mathcal{H}(\mathbf{Q})$ to Equ.(1) and solve the entropic regularized OT with the efficient scaling algorithm (Knight, 2008). Subsequently, Chizat et al. (2018) generalizes the scaling algorithm to solve the unbalanced OT problem. Additionally, one can introduce the total mass constraint to Equ.(1), and it can be solved by the efficient scaling algorithm by adding a dummy or virtual point to absorb the total mass constraint into marginal. We detail the scaling algorithm in Appendix A and refer readers to (Cuturi, 2013; Chizat et al., 2018) for more details.

## 4  METHOD

### 4.1  PROBLEM SETUP AND METHOD OVERVIEW

In deep imbalanced clustering, the training dataset is denoted as $\mathcal{D} = \{(x_i)\}_{i=1}^N$, where the cluster labels and distribution are unknown. The number of clusters $K$ is given as a prior. The goal is to learn representation and semantic clusters. To achieve that, we learn representation and pseudo-label alternately, improving the data representation and the quality of cluster assignments. Specifically, given the cluster prediction, we utilize our novel Progressive Partial Optimal Transport (P$^2$OT) algorithm to generate high-quality pseudo labels. Our P$^2$OT algorithm has two advantages: 1) generating pseudo labels in an imbalanced manner; 2) reweighting confident samples through optimization. Then, given the pseudo label, we update the representation. We alternate the above two steps until convergence. In the following section, we mainly detail our P$^2$OT algorithm.

### 4.2  PROGRESSIVE PARTIAL OPTIMAL TRANSPORT (P$^2$OT)

In this section, we derive the formulation of our novel P$^2$OT algorithm and illustrate how to infer pseudo label $\mathbf{Q}$ by the P$^2$OT algorithm. Given the model's prediction and its pseudo-label, the loss function is denoted as follows:

$$\mathcal{L} = -\sum_{i=1}^N \mathbf{Q}_i \log \mathbf{P}_i = \langle \mathbf{Q}, -\log \mathbf{P} \rangle_F, \tag{2}$$

where $\mathbf{Q}, \mathbf{P} \in \mathbb{R}_+^{N \times K}$, $\mathbf{Q}_i, \mathbf{P}_i$ is the pseudo label and prediction of sample $x_i$. Note that we have absorbed the normalize term $\frac{1}{N}$ into $\mathbf{Q}$ for simplicity, thus $\mathbf{Q}\mathbf{1}_K = \frac{1}{N}\mathbf{1}_N$.

In deep clustering, it is typical to impose some constraint on the cluster size distribution to avoid a degenerate solution, where all the samples are assigned to a single cluster. As the cluster distribution is

long-tailed and unknown, we adopt a $KL$ divergence constraint and only assume the prior distribution is uniform. Therefore, with two marginal distribution constraints, we can formulate the pseudo-label generation problem as an unbalanced OT problem:

$$\min_{\mathbf{Q} \in \Pi} \langle \mathbf{Q}, -\log \mathbf{P} \rangle_F + \lambda KL(\mathbf{Q}^\top \mathbf{1}_N, \frac{1}{K}\mathbf{1}_K) \tag{3}$$

$$\text{s.t.} \quad \Pi = \{\mathbf{Q} \in \mathbb{R}_+^{N \times K} | \mathbf{Q}\mathbf{1}_K = \frac{1}{N}\mathbf{1}_N\} \tag{4}$$

where $\langle,\rangle_F$ is the Frobenius product, and $\lambda$ is a scalar factor. In Equ.(3), the first term is exactly $\mathcal{L}$, the $KL$ term is a constraint on cluster size, and the equality term ensures each sample is equally important. However, the unbalanced OT algorithm treats each sample equally, which may generate noisy pseudo labels, due to the initial representation being poor, resulting in confirmation bias. Inspired by curriculum learning, which first learns from easy samples and gradually learns hard samples, we select only a fraction of high-confident samples to learn initially and increase the fraction gradually. However, instead of manually selecting confident samples through thresholding, we formulate the selection process as a total mass constraint in Equ.(5). This approach allows us to reweight each sample through joint optimization with the pseudo-label generation, eliminating the need for sensitive hyperparameter tuning. Therefore, the formulation of our novel P$^2$OT is as follows:

$$\min_{\mathbf{Q} \in \Pi} \langle \mathbf{Q}, -\log \mathbf{P} \rangle_F + \lambda KL(\mathbf{Q}^\top \mathbf{1}_N, \frac{\rho}{K}\mathbf{1}_K) \tag{5}$$

$$\text{s.t.} \quad \Pi = \{\mathbf{Q} \in \mathbb{R}_+^{N \times K} | \mathbf{Q}\mathbf{1}_K \le \frac{1}{N}\mathbf{1}_N, \mathbf{1}_N^\top \mathbf{Q}\mathbf{1}_K = \rho\} \tag{6}$$

where $\rho$ is the fraction of selected mass and will increase gradually, and $KL$ is the unnormalized divergence measure, enabling us to handle imbalance distribution. The resulted $\mathbf{Q}\mathbf{1}_K$ is the weight for samples. Intuitively, the P$^2$OT subsumes a set of prior distribution constraints, achieving imbalanced pseudo-label generation and confident sample "selection" in a single optimization problem.

**Increasing strategy of $\rho$.** In our formulation, the ramp-up strategy for the parameter $\rho$ plays an important role in model performance. Instead of starting with 0 and incrementally increasing to 1, we introduce an initial value $\rho_0$ to mitigate potential issues associated with the model learning from very limited samples in the initial stages. And we utilize the sigmoid ramp-up function, a technique commonly employed in semi-supervised learning (Samuli Laine, 2017; Tarvainen & Valpola, 2017). Therefore, the increasing strategy of $\rho$ is expressed as follows:

$$\rho = \rho_0 + (1 - \rho_0) \cdot e^{-5(1-t/T)^2}, \tag{7}$$

where $T$ represents the total number of iterations, and $t$ represents the current iteration. We analyze and compare alternate design choices (e.g. linear ramp-up function) in our ablation study. Furthermore, we believe that a more sensible approach to setting $\rho$ involves adaptively increasing it based on the model's learning progress rather than relying on a fixed parameterization tied to the number of iterations. We leave the advanced design of $\rho$ for future work.

### 4.3 Efficient Solver for P$^2$OT

In this section, we reformulate our P$^2$OT into an unbalanced OT problem and solve it with an efficient scaling algorithm. Our approach involves introducing a virtual cluster onto the marginal (Caffarelli & McCann, 2010; Chapel et al., 2020). This virtual cluster serves the purpose of absorbing the $1 - \rho$ unselected mass, enabling us to transform the total mass constraint into the marginal constraint. Additionally, we replace the $KL$ constraint with a weighted $KL$ constraint to ensure strict adherence to the total mass constraint. As a result, we can reformulate Equ.(5) into a form akin to Equ.(1) and prove their solutions can be interconverted. Subsequently, we resolve this reformulated problem using an efficient scaling algorithm. As shown in Sec.5.3, compared to the generalized scaling solver proposed by (Chizat et al., 2018), our solver is two times faster.

Specifically, we denote the assignment of samples on the virtual cluster as $\boldsymbol{\xi}$. Then, we extend $\boldsymbol{\xi}$ to $\mathbf{Q}$, and denote the extended $\mathbf{Q}$ as $\hat{\mathbf{Q}}$ which satisfies the following constraints:

$$\hat{\mathbf{Q}} = [\mathbf{Q}, \boldsymbol{\xi}] \in \mathbb{R}^{N \times (K+1)}, \quad \boldsymbol{\xi} \in \mathbb{R}^{N \times 1}, \quad \hat{\mathbf{Q}}\mathbf{1}_{K+1} = \mathbf{1}_N. \tag{8}$$

Due to $\mathbf{1}_N^\top \mathbf{Q}\mathbf{1}_K = \rho$, we known that,

$$\mathbf{1}_N^\top \hat{\mathbf{Q}}\mathbf{1}_{K+1} = \mathbf{1}_N^\top \mathbf{Q}\mathbf{1}_K + \mathbf{1}_N^\top \boldsymbol{\xi} = 1 \Rightarrow \mathbf{1}_N^\top \boldsymbol{\xi} = 1 - \rho. \tag{9}$$

Therefore,

$$\hat{\mathbf{Q}}^\top \mathbf{1}_N = \begin{bmatrix} \mathbf{Q}^\top \mathbf{1}_N \\ \boldsymbol{\xi}^\top \mathbf{1}_N \end{bmatrix} = \begin{bmatrix} \mathbf{Q}^\top \mathbf{1}_N \\ 1 - \rho \end{bmatrix}. \tag{10}$$

The equation is due to $1_N^\top \boldsymbol{\xi} = \boldsymbol{\xi}^T \mathbf{1}_N = 1 - \rho$. We denote $\mathbf{C} = [-\log \mathbf{P}, \mathbf{0}_N]$ and replace $\mathbf{Q}$ with $\hat{\mathbf{Q}}$, thus the Equ.(5) can be rewritten as follows:

$$\min_{\hat{\mathbf{Q}} \in \Phi} \langle \hat{\mathbf{Q}}, \mathbf{C} \rangle_F + \lambda KL(\hat{\mathbf{Q}}^\top \mathbf{1}_N, \boldsymbol{\beta}), \tag{11}$$

$$\text{s.t.} \quad \Phi = \{\hat{\mathbf{Q}} \in \mathbb{R}_+^{N \times (K+1)} | \hat{\mathbf{Q}} \mathbf{1}_{K+1} = \frac{1}{N} \mathbf{1}_N\}, \quad \boldsymbol{\beta} = \begin{bmatrix} \frac{\rho}{K} \mathbf{1}_K \\ 1 - \rho \end{bmatrix}. \tag{12}$$

However, the Equ.(11) is not equivalent to Equ.(5), due to the $KL$ constraint can not guarantee Equ.(10) is strictly satisfied, i.e. $\boldsymbol{\xi}^\top \mathbf{1}_N = 1 - \rho$. To solve this problem, we replace the $KL$ constraint with weighted $KL$, which enables us to control the constraint strength for each class. The formula of weighted $KL$ is denoted as follows:

$$\hat{KL}(\hat{\mathbf{Q}}^\top \mathbf{1}_N, \boldsymbol{\beta}, \boldsymbol{\lambda}) = \sum_{i=1}^{K+1} \boldsymbol{\lambda}_i [\hat{\mathbf{Q}}^\top \mathbf{1}_N]_i \log \frac{[\hat{\mathbf{Q}}^\top \mathbf{1}_N]_i}{\boldsymbol{\beta}_i}. \tag{13}$$

Therefore, the Equ.(5) can be rewritten as follows:

$$\min_{\hat{\mathbf{Q}} \in \Phi} \langle \hat{\mathbf{Q}}, \mathbf{C} \rangle_F + \hat{KL}(\hat{\mathbf{Q}}^\top \mathbf{1}_N, \boldsymbol{\beta}, \boldsymbol{\lambda}) \tag{14}$$

$$\text{s.t.} \quad \Phi = \{\hat{\mathbf{Q}} \in \mathbb{R}_+^{N \times (K+1)} | \hat{\mathbf{Q}} \mathbf{1}_{K+1} = \frac{1}{N} \mathbf{1}_N\}, \quad \boldsymbol{\beta} = \begin{bmatrix} \frac{\rho}{K} \mathbf{1}_K \\ 1 - \rho \end{bmatrix}, \quad \boldsymbol{\lambda}_{K+1} \to +\infty. \tag{15}$$

Intuitively, to assure Equ.(10) is strictly satisfied, we set $\boldsymbol{\lambda}_{K+1} \to +\infty$. This places a substantial penalty on the virtual cluster, compelling the algorithm to assign a size of $1 - \rho$ to the virtual cluster.

**Proposition 1** *If $\mathbf{C} = [-\log \mathbf{P}, \mathbf{0}_N]$, and $\boldsymbol{\lambda}_{:K} = \lambda, \boldsymbol{\lambda}_{K+1} \to +\infty$, the optimal transport plan $\hat{\mathbf{Q}}^\star$ of Equ.(14) can be expressed as:*

$$\hat{\mathbf{Q}}^\star = [\mathbf{Q}^\star, \boldsymbol{\xi}^\star], \tag{16}$$

*where $\mathbf{Q}^\star$ is optimal transport plan of Equ.(5), and $\boldsymbol{\xi}^\star$ is the last column of $\hat{\mathbf{Q}}^\star$.*

The proof is in Appendix B. Consequently, we focus on solving Equ.(14) to obtain the optimal $\hat{\mathbf{Q}}^\star$.

**Proposition 2** *Adding a entropy regularization $-\epsilon \mathcal{H}(\hat{\mathbf{Q}})$ to Equ.(14), we can solve it by efficient scaling algorithm. We denote $\mathbf{M} = \exp(-\mathbf{C}/\epsilon), \boldsymbol{f} = \frac{\boldsymbol{\lambda}}{\boldsymbol{\lambda}+\epsilon}, \boldsymbol{\alpha} = \frac{1}{N} \mathbf{1}_N$. The optimal $\hat{\mathbf{Q}}^\star$ is denoted as follows:*

$$\hat{\mathbf{Q}}^\star = \text{diag}(\mathbf{a}) \mathbf{M} \text{diag}(\mathbf{b}), \tag{17}$$

*where $\mathbf{a}, \mathbf{b}$ are two scaling coefficient vectors and can be derived by the following recursion formula:*

$$\mathbf{a} \leftarrow \frac{\boldsymbol{\alpha}}{\mathbf{M}\mathbf{b}}, \quad \mathbf{b} \leftarrow (\frac{\boldsymbol{\beta}}{\mathbf{M}^\top \mathbf{a}})^{\circ \boldsymbol{f}}, \tag{18}$$

*where $\circ$ denotes Hadamard power, i.e., element-wise power. The recursion will stop until $\boldsymbol{b}$ converges.*

The proof is in Appendix C, and the pseudo-code is shown in Alg.1. The efficiency analysis is in the Sec.5.3. In practice, rather than solving the P$^2$OT for the entire dataset, we implement a mini-batch approach and store historical predictions as a memory buffer to stabilize optimization.

## 5 EXPERIMENTS

### 5.1 EXPERIMENTAL SETUP

**Datasets.** To evaluate our method, we have established a realistic and challenging benchmark, including CIFAR100 (Krizhevsky et al., 2009), ImageNet-R (abbreviated as ImgNet-R) (Hendrycks et al., 2021) and iNaturalist2018 (Van Horn et al., 2018) datasets. To quantify the level of class imbalance, we introduce the imbalance ratio denoted as $R$, calculated as the ratio of $N_{max}$ to $N_{min}$,

---

**Algorithm 1:** Scaling Algorithm for P$^2$OT

---

**Input:** Cost matrix $-\log\mathbf{P}$, $\epsilon$, $\lambda$, $\rho$, $N$, $K$, a large value $\iota$
$\mathbf{C} \leftarrow [-\log\mathbf{P}, \mathbf{0}_N], \quad \boldsymbol{\lambda} \leftarrow [\lambda, ..., \lambda, \iota]^\top$
$\boldsymbol{\beta} \leftarrow [\frac{\rho}{K}\mathbf{1}_K^\top, 1-\rho]^\top, \quad \boldsymbol{\alpha} \leftarrow \frac{1}{N}\mathbf{1}_N$
$\mathbf{b} \leftarrow \mathbf{1}_{K+1}, \quad \mathbf{M} \leftarrow \exp(-\mathbf{C}/\epsilon), \quad \boldsymbol{f} \leftarrow \frac{\lambda}{\lambda+\epsilon}$
**while** $\mathbf{b}$ *not converge* **do**
    $\mathbf{a} \leftarrow \frac{\boldsymbol{\alpha}}{\mathbf{Mb}}$
    $\mathbf{b} \leftarrow (\frac{\boldsymbol{\beta}}{\mathbf{M}^\top\mathbf{a}})^{\circ\boldsymbol{f}}$
**end**
$\mathbf{Q} \leftarrow \mathrm{diag}(\mathbf{a})\mathbf{M}\mathrm{diag}(\mathbf{b})$
**return** $\mathbf{Q}[:, :K]$

---

where $N_{max}$ represents the largest number of images in a class, and $N_{min}$ represents the smallest. For CIFAR100, as in (Cao et al., 2019), we artificially construct a long-tailed CIFAR100 dataset with an imbalance ratio of 100. For ImgNet-R, which has renditions of 200 classes resulting in 30k images and is inherently imbalanced, we split 20 images per class as the test set, leaving the remaining data as the training set ($R = 13$). Note that the data distribution of ImgNet-R is different from the ImageNet, which is commonly used for training unsupervised pre-trained models, posing a great challenge to its clustering. Consequently, ImgNet-R serves as a valuable resource for assessing the robustness of various methods. Furthermore, we incorporate the iNaturalist2018 dataset, a natural long-tailed dataset frequently used in supervised long-tailed learning (Cao et al., 2019; Zhang et al., 2021b). This dataset encompasses 8,142 classes, posing significant challenges for clustering. To mitigate this complexity, we extract subsets of 100, 500, and 1000 classes, creating the iNature100 ($R = 67$), iNature500 ($R = 111$), and iNature1000 ($R = 111$) datasets, respectively. iNature100 is the subset of iNature500, and iNature500 is the subset of iNature1000. The distribution of datasets is in Appendix E. We perform evaluations on both the imbalanced training set and the corresponding balanced test set. Note that we do not conduct experiments on ImageNet datasets because the unsupervised pretrained models have trained on the whole balanced ImageNet.

**Evaluation Metric.** We evaluate our method using the clustering accuracy (ACC) metric averaged over classes, normalized mutual information (NMI), and F1-score. We also provide the adjusted Rand index (ARI) metrics, which is not a suitable metric for imbalanced datasets, in Appendix H. To provide a more detailed analysis, we rank the classes by size in descending order and divide the dataset into Head, Medium, and Tail categories, maintaining a ratio of 3:4:3 across all datasets. Then, we evaluate performance on the Head, Medium, and Tail, respectively.

**Implementation Details.** Building upon the advancements in transformer (Dosovitskiy et al., 2020) and unsupervised pre-trained models (He et al., 2022), we conduct experiments on the ViT-B16, which is pre-trained with DINO (Caron et al., 2021). To provide a comprehensive evaluation, we re-implement most methods from the existing literature, including the typical IIC (Ji et al., 2019), PICA (Huang et al., 2020), CC (Li et al., 2021), SCAN (Van Gansbeke et al., 2020), strong two-stage based SCAN* (Van Gansbeke et al., 2020) and SPICE (Niu et al., 2022), and the recently proposed DivClust (Metaxas et al., 2023). In addition, we also implement BCL (Zhou et al., 2022b), which is specifically designed for representation learning with long-tailed data. It is important to note that all of these methods are trained using the same backbone, data augmentation, and training configurations to ensure a fair comparison. Specifically, we train 50 epochs and adopt the Adam optimizer with the learning rate decay from 5e-4 to 5e-6 for all datasets. The batch size is 512. Further details can be found in Appendix F. For hyperparameters, we set $\lambda$ as 1, $\epsilon$ as 0.1, and initial $\rho$ as 0.1. The stop criterion of Alg.1 is when the change of $\mathbf{b}$ is less than 1e-6, or the iteration reaches 1000. We utilize the loss on training sets for clustering head and model selection. For evaluation, we conduct experiments with each method three times and report the mean results.

## 5.2 MAIN RESULTS

In Tab.1, we provide a comprehensive comparison of our method with existing approaches on various imbalanced training sets. The results for iNature1000 and balanced test sets can be found in Appendix H. On the relatively small-scale CIFAR100 dataset, our method outperforms the previous state-of-the-art by achieving an increase of 0.9 in ACC, 0.3 in NMI, and 0.6 in F1 score. On the ImgNet-R datasets, our method demonstrates its effectiveness with significant improvements of 2.4

Table 1: Comparison with SOTA methods on different imbalanced training sets. The best results are shown in boldface, and the next best results are indicated with an underscore.

| Method | CIFAR100 | | | ImgNet-R | | | iNature100 | | | iNature500 | | |
|---|---|---|---|---|---|---|---|---|---|---|---|---|
| | ACC | NMI | F1 | ACC | NMI | F1 | ACC | NMI | F1 | ACC | NMI | F1 |
| DINO | 36.6 | 68.9 | 31.0 | 20.5 | 39.6 | 22.2 | 40.1 | 67.8 | 34.2 | 29.8 | 67.0 | 24.0 |
| BCL | 35.7 | 66.0 | 29.9 | 20.7 | 40.0 | 22.4 | 41.9 | 67.2 | 35.4 | 28.1 | 64.7 | 22.4 |
| IIC | $27.3_{\pm3.1}$ | $65.0_{\pm1.8}$ | $23.0_{\pm2.6}$ | $18.7_{\pm1.5}$ | $39.6_{\pm1.1}$ | $15.9_{\pm1.5}$ | $28.5_{\pm1.6}$ | $63.9_{\pm1.0}$ | $22.2_{\pm1.2}$ | $13.1_{\pm0.3}$ | $58.4_{\pm0.3}$ | $7.1_{\pm0.2}$ |
| PICA | $29.8_{\pm0.6}$ | $59.9_{\pm0.2}$ | $24.0_{\pm0.2}$ | $12.6_{\pm0.3}$ | $34.0_{\pm0.0}$ | $12.1_{\pm0.2}$ | $34.8_{\pm2.4}$ | $54.8_{\pm0.6}$ | $23.8_{\pm1.2}$ | $16.3_{\pm0.3}$ | $55.9_{\pm0.1}$ | $11.3_{\pm0.1}$ |
| SCAN | $\underline{37.2}_{\pm0.9}$ | $\underline{69.4}_{\pm0.4}$ | $\underline{31.4}_{\pm0.7}$ | $21.8_{\pm0.7}$ | $42.6_{\pm0.3}$ | $21.7_{\pm0.8}$ | $38.7_{\pm0.6}$ | $66.3_{\pm0.5}$ | $28.4_{\pm0.6}$ | $\underline{29.0}_{\pm0.3}$ | $\underline{66.7}_{\pm0.2}$ | $21.6_{\pm0.2}$ |
| SCAN* | $30.2_{\pm0.9}$ | $68.5_{\pm2.3}$ | $25.4_{\pm1.1}$ | $\underline{23.6}_{\pm0.2}$ | $\underline{44.1}_{\pm0.2}$ | $\underline{22.8}_{\pm0.1}$ | $\underline{39.5}_{\pm0.4}$ | $\mathbf{68.5}_{\pm0.2}$ | $30.7_{\pm0.1}$ | $19.0_{\pm0.7}$ | $65.9_{\pm0.5}$ | $12.5_{\pm0.5}$ |
| CC | $29.0_{\pm0.6}$ | $60.7_{\pm0.6}$ | $24.6_{\pm0.5}$ | $12.1_{\pm0.6}$ | $30.5_{\pm0.1}$ | $11.2_{\pm0.9}$ | $28.2_{\pm2.4}$ | $56.1_{\pm1.2}$ | $20.6_{\pm2.1}$ | $16.5_{\pm0.7}$ | $55.5_{\pm0.0}$ | $12.3_{\pm0.4}$ |
| DivClust | $31.8_{\pm0.3}$ | $64.0_{\pm0.4}$ | $26.1_{\pm0.8}$ | $14.8_{\pm0.2}$ | $33.9_{\pm0.4}$ | $13.8_{\pm0.2}$ | $33.7_{\pm0.2}$ | $59.3_{\pm0.5}$ | $23.3_{\pm0.7}$ | $17.2_{\pm0.5}$ | $56.4_{\pm0.3}$ | $12.5_{\pm0.2}$ |
| P²OT | $\mathbf{38.2}_{\pm0.8}$ | $\mathbf{69.6}_{\pm0.3}$ | $\mathbf{32.0}_{\pm0.9}$ | $\mathbf{25.9}_{\pm0.9}$ | $\mathbf{45.7}_{\pm0.5}$ | $\mathbf{27.3}_{\pm1.4}$ | $\mathbf{44.2}_{\pm1.2}$ | $\underline{67.0}_{\pm0.6}$ | $\mathbf{36.9}_{\pm2.0}$ | $\mathbf{32.2}_{\pm2.0}$ | $\mathbf{67.2}_{\pm0.3}$ | $\mathbf{25.2}_{\pm1.7}$ |

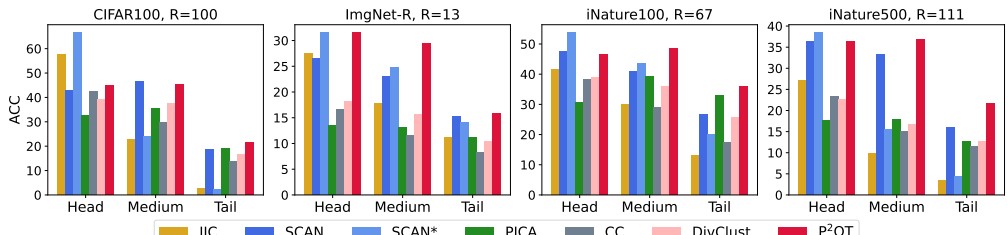

Figure 1: Head, Medium, and Tail comparison on several datasets.

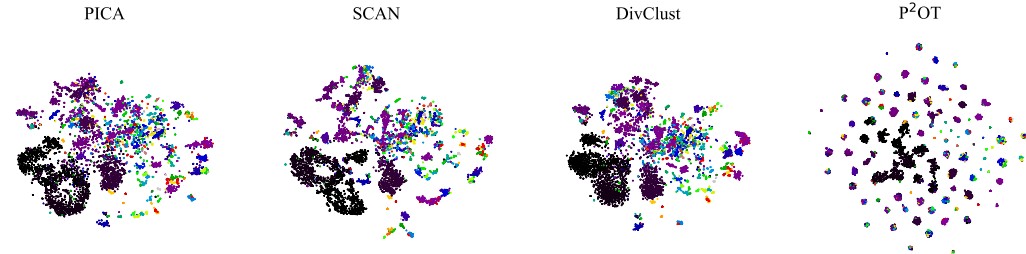

Figure 2: The T-SNE analysis on iNature100 training set.

in ACC, 1.6 in NMI, and 4.5 in F1 score, highlighting its robustness in out-of-distribution scenarios. When applied to the fine-grained iNature datasets, our approach consistently delivers substantial performance gains across each subset in terms of ACC and F1 scores. Specifically, on ACC, we achieve improvements of 5.9 on iNature100 and 3.4 on iNature500. On the F1 score, we obtain improvements of 1.5 and 1.2 on the two datasets, respectively. In terms of NMI, we observe a 0.6 improvement on iNature500 but a 1.5 decrease on iNature100. It's worth noting that our method is an efficient one-stage approach, unlike SCAN*, which is a two-stage pseudo-labeling-based method. Additionally, another pseudo-labeling-based method, SPICE, exhibits a degenerate solution in the imbalance scenario (see Appendix F). Those results indicate the naive pseudo-labeling methods encounter a great challenge and demonstrate our superiority in handling imbalance scenarios.

In addition, we provide a detailed analysis of the results for the Head, Medium, and Tail classes, offering a more comprehensive understanding of our method's performance across different class sizes. As depicted in Fig. 1, our improvements are predominantly driven by the Medium and Tail classes, especially in challenging scenarios like ImgNet-R and iNature500, although our results show some reductions in performance for the Head classes. These results highlight the effectiveness of our P²OT algorithm in generating imbalance-aware pseudo-labels, making it particularly advantageous for the Medium and Tail classes. Furthermore, in Fig.2, we present a T-SNE (Van der Maaten & Hinton, 2008) comparison of features before the clustering head. The T-SNE results illustrate our method learns more distinct clusters, particularly benefiting Medium and Tail classes.

## 5.3 ABLATION STUDY

**Component Analysis.** To assess the $KL$ constraint and the progressive $\rho$, we conduct ablation experiments to analyze their individual contributions. As shown in Tab.2, we compare our full P²OT method with three ablated versions: OT, POT and UOT. OT (Asano et al., 2020) imposes equality constraint on cluster size and without the progressive $\rho$ component. POT signifies the replacement of

Table 2: Analysis of different formulations. SLA is proposed by (Tai et al., 2021) in semi-supervised learning. OT (Asano et al., 2020), POT and UOT are variants of our P$^2$OT. POT substitutes the $KL$ constraint with the equality constraint. UOT removes the progressive $\rho$.

| Formulation | CIFAR100 | | | | ImgNet-R | | | | iNature500 | | | |
|---|---|---|---|---|---|---|---|---|---|---|---|---|
| | Head | Medium | Tail | ACC | Head | Medium | Tail | ACC | Head | Medium | Tail | ACC |
| SLA | - | - | - | 3.26 | - | - | - | 1.00 | - | - | - | 6.33 |
| OT | 35.0 | 30.2 | 18.7 | 28.2 | 24.4 | 19.3 | 10.7 | 18.2 | 27.1 | 27.5 | 16.4 | 24.1 |
| POT | 39.6 | 42.9 | 14.5 | 33.4 | 25.5 | 23.3 | 17.6 | 22.2 | 31.7 | 32.7 | 19.7 | 28.5 |
| UOT | 43.4 | 42.5 | 19.6 | 35.9 | 27.7 | 23.9 | 14.6 | 22.2 | 32.2 | 29.0 | 18.3 | 26.7 |
| P$^2$OT | 45.1 | 45.5 | 21.6 | 38.2 | 31.5 | 29.3 | 16.1 | 25.9 | 36.3 | 37.0 | 21.6 | 32.2 |

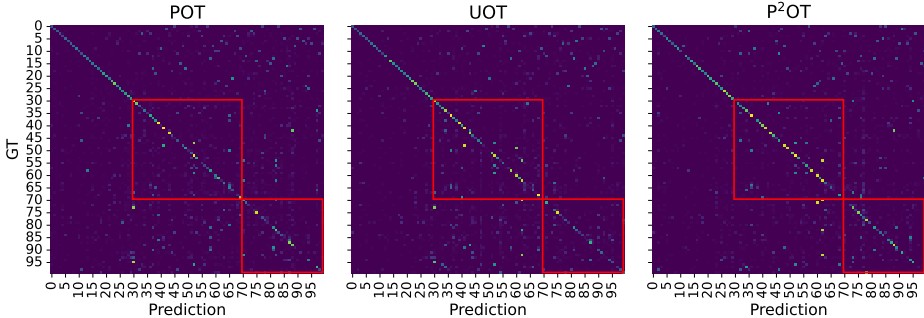

Figure 3: Confusion matrix on the balanced CIFAR100 test set. The two red rectangles represent the Medium and Tail classes.

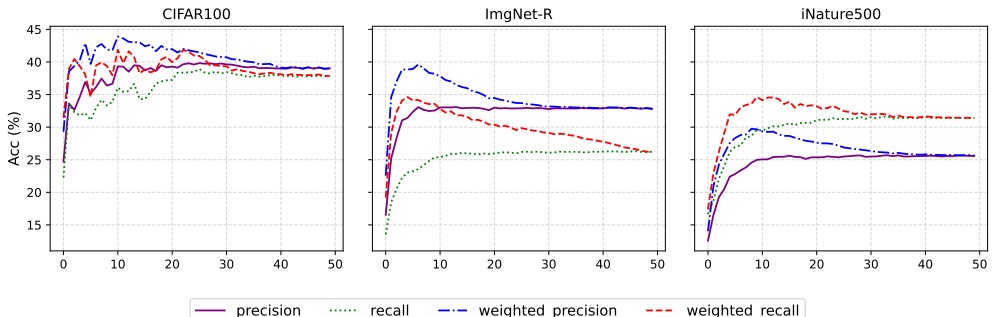

Figure 4: Precision, Recall analysis on train dataset with different training epoch. The weighted precision and recall are derived by reweighting each sample by our P$^2$OT algorithm.

the $KL$ term in the original P$^2$OT formulation with a typical uniform constraint. UOT denotes the removal of the progressive $\rho$ component from the original P$^2$OT formulation. Furthermore, we also compare with SLA (Tai et al., 2021), which relaxes the equality constraint of POT by utilizing an upper bound. However, due to this relaxation, samples are erroneously assigned to a single cluster in the early stages, rendering its failure for clustering imbalanced data. We detail their formulation and further analysis in Appendix J. The results show POT and UOT both significantly surpass the OT, indicating that both of our components yield satisfactory effects. Compared to POT, we achieve improvements of 5.8, 3.8, and 3.6 on CIFAR100, ImgNet-R, and iNature500, respectively. Our improvement is mainly from Head and Tail classes, demonstrating that our P2OT with $KL$ constraint can generate imbalanced pseudo labels. Compared to UOT, we realize gains of 2.3, 3.8, and 5.4 on CIFAR100, ImgNet-R, and iNature500, respectively. Furthermore, as depicted in Fig.3, the confusion matrix for the CIFAR100 test set also validates our improvement on Medium and Tail classes. These remarkable results underscore the effectiveness of each component in our P$^2$OT.

**Pseudo Label Quality Analysis.** To provide a deeper understanding, we conduct an analysis of the pseudo-label quality generated by our P$^2$OT. We evaluate precision and recall metrics to assess the pseudo-label quality for the entire training set. Notably, our P$^2$OT algorithm conducts selection through reweighting, rather than hard selection. Consequently, we reweight each sample and present the weighted precision and recall results. As depicted in Fig.4, our weighted precision and recall consistently outperform precision and recall across different epochs. In the early stages, weighted precision and recall exhibit more rapid improvements compared to precision and recall. However,

Table 3: Analysis of $\rho_0$ and different ramp up function. Fixed denotes $\rho$ as a constant value.

| $\rho_0$ | Sigmoid | | | | | Linear | | | Fixed | |
|---|---|---|---|---|---|---|---|---|---|---|
| | 0.00 | 0.05 | 0.1 | 0.15 | 0.2 | 0 | 0.1 | 0.2 | 0.1 | 0.2 |
| CIFAR100 | 32.9 | 37.9 | 38.2 | 36.3 | 37.6 | 35.0 | 36.6 | 36.8 | 37.4 | 37.1 |
| ImgNet-R | 26.5 | 27.2 | 25.9 | 26.0 | 26.6 | 25.8 | 28.5 | 23.0 | 25.5 | 24.7 |
| iNature500 | 32.6 | 31.7 | 32.2 | 32.8 | 32.9 | 31.9 | 32.0 | 30.5 | 29.9 | 30.3 |

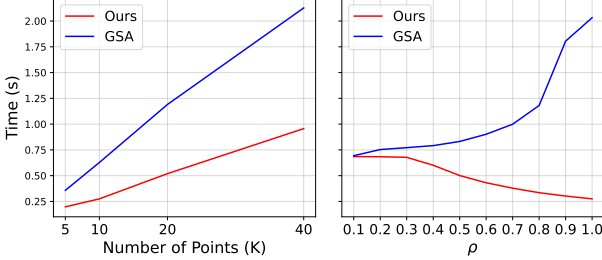

Figure 5: Time cost comparison of our solver with Generalized Scaling Algorithm (GSA).

they eventually plateau around 10 epochs ($\rho \approx 0.15$), converging gradually with precision and recall. The decline observed in weighted precision and recall over time suggests that our current ramp-up function may not be optimal, and raising to 1 for $\rho$ may not be necessary. We believe that the ramp-up strategy for $\rho$ should be adaptive to the model's learning progress. In this paper, we have adopted a typical sigmoid ramp-up strategy, and leave more advanced designs for future work.

**Analysis of $\rho$.**   In our P²OT algorithm, the choice of initial $\rho_0$ and the specific ramp-up strategy are important hyperparameters. In this section, we systematically investigate the impact of varying $\rho_0$ values and alternative ramp-up strategy. The term "Linear" signifies that $\rho$ is increased to 1 from $\rho_0$ using a linear function, while "Fixed" indicates that $\rho$ remains constant as $\rho_0$. The results in Tab.3 provide several important insights: 1) Our method exhibits consistent performance across various $\rho_0$ values when using the sigmoid ramp-up function, highlighting its robustness on datasets like ImgNet-R and iNature500; 2) The linear ramp-up strategy, although slightly less effective than sigmoid, still demonstrates the importance of gradually increasing $\rho$ during the early training stage; 3) The fixed $\rho$ approach results in suboptimal performance, underscoring the necessity of having a dynamically increasing $\rho$. These findings suggest that a good starting value for $\rho_0$ is around 0.1, and it should be progressively increased during training.

**Efficiency Analysis.**   To demonstrate the efficiency of our solver, we perform a comparison between our solver and the Generalized Scaling Algorithm (GSA) proposed by (Chizat et al., 2018). The pseudo-code of GSA is in Appendix K. This comparison is conducted on iNaure1000 using identical conditions (NVIDIA TITAN RTX, $\epsilon = 0.1, \lambda = 1$), without employing any acceleration strategies for both. To ensure the comprehensiveness of our time cost analysis, we conduct experiments by varying both the number of data points and the $\rho$ value. Subsequently, we average the time costs for different numbers of data points across different $\rho$ values and vice versa. The results presented in Fig.5 reveal several insights: 1) the time cost of both methods increases near linearly with the number of points; 2) as $\rho$ approaches 1, the time cost of GSA increases rapidly due to the tightening inequality constraint, whereas our time cost decreases; 3) as shown in the left figure, our solver is 2× faster than GSA. Those results demonstrate the efficiency of our solver.

## 6   CONCLUSION

In this paper, we introduce a more practical problem called "deep imbalanced clustering," which aims to learn representations and semantic clusters from imbalanced data. To address this challenging problem, we propose a novel progressive PL-based learning framework, which formulates the pseudo-label generation as a progressive partial optimal transport (P²OT) algorithm. The P²OT enables us to generate high-quality pseudo-labels in an imbalance-aware and progressive manner, thereby significantly improving model learning and clustering performance. To efficiently solve the P²OT algorithm, we introduce a virtual cluster and a weighted $KL$ constraint. Subsequently, by imposing certain constraints, we transform our problem into an unbalanced optimal transport problem, which can be efficiently solved using a scaling algorithm. Finally, extensive results on the human-curated long-tailed CIFAR100 dataset, challenging ImageNet-R datasets, and several large-scale fine-grained iNature datasets validate the superiority of our method.

ACKNOWLEDGMENTS

This work was supported by National Science Foundation of China under grant 62350610269, Shanghai Frontiers Science Center of Human-centered Artificial Intelligence, and MoE Key Lab of Intelligent Perception and Human-Machine Collaboration (ShanghaiTech University). We thank Wanxing Chang for her valuable and inspiring comments.

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

## A  Efficient Scaling Algorithm for solving Optimal Transport

In this section, we detail how to solve the optimal transport by an efficient scaling algorithm. Let's recall the definition of optimal transport, given two probability vectors $\boldsymbol{\mu} \in \mathbb{R}^{m \times 1}, \boldsymbol{\nu} \in \mathbb{R}^{n \times 1}$, as well as a cost matrix $\mathbf{C} \in \mathbb{R}_+^{m \times n}$ defined on joint space, the objective function which OT minimizes is as follows:

$$\min_{\mathbf{Q} \in \mathbb{R}_+^{m \times n}} \langle \mathbf{Q}, \mathbf{C} \rangle_F + F_1(\mathbf{Q}\mathbf{1}_n, \boldsymbol{\mu}) + F_2(\mathbf{Q}^\top \mathbf{1}_m, \boldsymbol{\nu}) \tag{S19}$$

where $\mathbf{Q} \in \mathbb{R}_+^{m \times n}$ is the transportation plan, $\langle , \rangle_F$ denotes the Frobenius product, $F_1, F_2$ are constraints on the marginal distribution of $\mathbf{Q}$, which are convex, lower semicontinuous and lower bounded functions, and $\mathbf{1}_n \in \mathbb{R}^{n \times 1}, \mathbf{1}_m \in \mathbb{R}^{m \times 1}$ are all ones vector. To solve it efficiently, motivated by Cuturi (2013), we first introduce an entropy constraint, $-\epsilon \mathcal{H}(\mathbf{Q})$. Therefore, the first term of Equ.(S19) is as follows:

$$<\mathbf{Q}, \mathbf{C}>_F - \epsilon \mathcal{H}(\mathbf{Q}) = \epsilon < \mathbf{Q}, \mathbf{C}/\epsilon + \log \mathbf{Q} >_F \tag{S20}$$

$$= \epsilon < \mathbf{Q}, \log \frac{\mathbf{Q}}{\exp(-\mathbf{C}/\epsilon)} >_F \tag{S21}$$

$$= \epsilon KL(\mathbf{Q}, \exp(-\mathbf{C}/\epsilon)), \tag{S22}$$

The entropic optimal transport can be reformulated as follows:

$$\min_{\mathbf{Q} \in \mathbb{R}_+^{m \times n}} \epsilon KL(\mathbf{Q}, \exp(-\mathbf{C}/\epsilon)) + F_1(\mathbf{Q}\mathbf{1}_n, \boldsymbol{\mu}) + F_2(\mathbf{Q}^\top \mathbf{1}_m, \boldsymbol{\nu}) \tag{S23}$$

Then, this problem can be approximately solved by an efficient scaling Alg.2, where the proximal operator is as follows:

$$\mathrm{prox}_{F/\epsilon}^{KL}(\mathbf{z}, \boldsymbol{\mu}) = \mathrm{argmin}_{\mathbf{x} \in \mathbb{R}_+^n} F(\mathbf{x}, \boldsymbol{\mu}) + \epsilon KL(\mathbf{x}, \mathbf{z}) \tag{S24}$$

Intuitively, it is the iterative projection on affine subspaces for the KL divergence. We refer readers to (Chizat et al., 2018) for more derivation. Consequently, for OT problems with any proper constraint, if we can transform it into the form of Equ.(S19) and derive the corresponding proximal operator, we can solve it with Alg.2 efficiently.

---

**Algorithm 2:** Scaling Algorithm for Optimal Transport

---

**Input:** Cost matrix $\mathbf{C}, \epsilon, m, n, \boldsymbol{\mu}, \boldsymbol{\nu}$
$\mathbf{M} = \exp(-\mathbf{C}/\epsilon)$
$\mathbf{b} \leftarrow \mathbf{1}_n$
**while** $b$ *not converge* **do**
  $\mathbf{a} \leftarrow \mathrm{prox}_{F_1/\epsilon}^{KL}(\mathbf{Mb}, \boldsymbol{\mu})/(\mathbf{Mb})$
  $\mathbf{b} \leftarrow \mathrm{prox}_{F_2/\epsilon}^{KL}(\mathbf{M}^\top \mathbf{a}, \boldsymbol{\nu})/(\mathbf{M}^\top \mathbf{a})$
**end**
**return** $\mathrm{diag}(\mathbf{a})\mathbf{M}\mathrm{diag}(\mathbf{b})$;

---

## B  Proof of Proposition 1

In this section, we prove the Proposition 1. Assume the optimal transportation plan of Equ.(14) is $\hat{\mathbf{Q}}^\star$, which can be decomposed as $[\tilde{\mathbf{Q}}^\star, \boldsymbol{\xi}^\star]$, and the optimal transportation plan of Equ.(5) is $\mathbf{Q}^\star$.

**Step 1.**  We first prove $\boldsymbol{\xi}^{\star\top} \mathbf{1}_N = 1 - \rho$. To prove that, we expand the second $\hat{KL}$ term and rewrite it as follows:

$$\hat{KL}(\hat{\mathbf{Q}}^{\star\top}\mathbf{1}_N, \boldsymbol{\beta}, \boldsymbol{\lambda}) = \sum_{i=1}^K \boldsymbol{\lambda}_i [\tilde{\mathbf{Q}}^{\star\top}\mathbf{1}_N]_i \log \frac{[\tilde{\mathbf{Q}}^{\star\top}\mathbf{1}_N]_i}{\boldsymbol{\beta}_i} + \boldsymbol{\lambda}_{K+1} \boldsymbol{\xi}^{\star\top}\mathbf{1}_N \log \frac{\boldsymbol{\xi}^{\star\top}\mathbf{1}_N}{1-\rho} \tag{S25}$$

$$= \lambda KL(\tilde{\mathbf{Q}}^{\star\top}\mathbf{1}_N, \frac{\rho}{K}\mathbf{1}_K) + \boldsymbol{\lambda}_{K+1}\boldsymbol{\xi}^{\star\top}\mathbf{1}_N \log \frac{\boldsymbol{\xi}^{\star\top}\mathbf{1}_N}{1-\rho} \tag{S26}$$

Due to $\boldsymbol{\lambda}_{K+1} \to +\infty$, if $\boldsymbol{\xi}^{\star\top}\mathbf{1}_N$ is not equal to $1-\rho$, the cost of Equ.(14) will be $+\infty$. In such case, we can find a more optimal $\hat{\mathbf{Q}}^\dagger$, which satisfies $\boldsymbol{\xi}^{\dagger\top}\mathbf{1}_N = 1-\rho$, and its cost is finite, contradicting to our assumption. Therefore, $\boldsymbol{\xi}^{\star\top}\mathbf{1}_N = 1-\rho$, and $\hat{KL}(\hat{\mathbf{Q}}^{\star\top}\mathbf{1}_N, \boldsymbol{\beta}, \boldsymbol{\lambda}) = \lambda KL(\tilde{\mathbf{Q}}^{\star\top}\mathbf{1}_N, \frac{\rho}{K}\mathbf{1}_K)$.

**Step 2.** We then prove $\tilde{\mathbf{Q}}^\star, \mathbf{Q}^\star$ are in the same constraint set. Due to

$$\hat{\mathbf{Q}}^\star\mathbf{1}_{K+1} = [\tilde{\mathbf{Q}}^\star, \boldsymbol{\xi}^\star]\mathbf{1}_{K+1} = \tilde{\mathbf{Q}}^\star\mathbf{1}_K + \boldsymbol{\xi}^\star = \mathbf{1}_N, \tag{S27}$$

and $\boldsymbol{\xi}^\star \geq 0$, we know $\tilde{\mathbf{Q}}^\star\mathbf{1}_K = \mathbf{1}_N - \boldsymbol{\xi}^\star \leq \mathbf{1}_N$. Furthermore,

$$\mathbf{1}_N^\top\hat{\mathbf{Q}}^\star\mathbf{1}_{K+1} = \mathbf{1}_N^\top(\tilde{\mathbf{Q}}^\star\mathbf{1}_K + \boldsymbol{\xi}^\star) = \mathbf{1}_N^\top\tilde{\mathbf{Q}}^\star\mathbf{1}_K + \mathbf{1}_N^\top\boldsymbol{\xi}^\star = 1, \tag{S28}$$

and $\boldsymbol{\xi}^{\star\top}\mathbf{1}_N = \mathbf{1}_N^\top\boldsymbol{\xi}^\star = 1-\rho$, we derive that

$$\mathbf{1}_N^\top\tilde{\mathbf{Q}}^\star\mathbf{1}_K = 1 - \mathbf{1}_N^\top\boldsymbol{\xi}^\star = 1 - (1-\rho) = \rho. \tag{S29}$$

In summary, $\tilde{\mathbf{Q}}^\star \in \{\tilde{\mathbf{Q}}^\star \in \mathbb{R}^{N \times K} | \tilde{\mathbf{Q}}^\star\mathbf{1}_K \leq \frac{1}{N}\mathbf{1}_N, \mathbf{1}_N^\top\tilde{\mathbf{Q}}^\star\mathbf{1}_K = \rho\}$, which is the same $\Pi$ in Equ.(5).

**Step 3.** Finally, we prove $\tilde{\mathbf{Q}}^\star = \mathbf{Q}^\star$. According to Proposition 1, $\mathbf{C} = [-\log\mathbf{P}, \mathbf{0}_N]$. We plug it into Equ.(14), and the cost of $\hat{\mathbf{Q}}^\star$ is as follows:

$$\langle\hat{\mathbf{Q}}^\star, \mathbf{C}\rangle_F + \hat{KL}(\hat{\mathbf{Q}}^{\star\top}\mathbf{1}_N, \boldsymbol{\beta}, \boldsymbol{\lambda}) = \langle[\tilde{\mathbf{Q}}^\star, \boldsymbol{\xi}^\star], [-\log\mathbf{P}, \mathbf{0}_N]\rangle_F + \lambda KL(\tilde{\mathbf{Q}}^{\star\top}\mathbf{1}_N, \frac{\rho}{K}\mathbf{1}_K) \tag{S30}$$

$$= \langle\tilde{\mathbf{Q}}^\star, -\log\mathbf{P}\rangle_F + \lambda KL(\tilde{\mathbf{Q}}^{\star\top}\mathbf{1}_N, \frac{\rho}{K}\mathbf{1}_K) \tag{S31}$$

$$= C_1 \tag{S32}$$

And in Equ.(5), the cost for $\mathbf{Q}^\star$ is as follows:

$$\langle\mathbf{Q}^\star, -\log\mathbf{P}\rangle_F + \lambda KL(\mathbf{Q}^{\star\top}\mathbf{1}_N, \frac{\rho}{K}\mathbf{1}_K) = C_2 \tag{S33}$$

From step 2, we know that $\tilde{\mathbf{Q}}^\star, \mathbf{Q}^\star$ are in the same set. Consequently, the form of Equ.(S30) is the same as Equ.(S33), and in set $\Pi$, $\langle\mathbf{Q}, -\log\mathbf{P}\rangle_F + \lambda KL(\mathbf{Q}^\top\mathbf{1}_N, \frac{\rho}{K}\mathbf{1}_K)$ is a convex function.

If $C_1 = C_2$, due to it is a convex function, $\tilde{\mathbf{Q}}^\star = \mathbf{Q}^\star$.

If $C_1 > C_2$, we can construct a transportation plan $\hat{\mathbf{Q}}^\dagger = [\mathbf{Q}^\star, \boldsymbol{\xi}^\star]$ for Equ.(14), which cost is $C_2$, achieving smaller cost than $\hat{\mathbf{Q}}^\star$. The results contradict the initial assumption that $\hat{\mathbf{Q}}^\star$ is the optimal transport plan. Therefore, $C_1$ can not be larger than $C_2$.

If $C_1 < C_2$, we can construct a transportation plan $\tilde{\mathbf{Q}}^\star$ for Equ.(5), which cost is $C_1$, achieving smaller cost than $\mathbf{Q}^\star$. The results contradict the initial assumption that $\mathbf{Q}^\star$ is the optimal transport plan. Therefore, $C_1$ can not be smaller than $C_2$.

In conclusion, $C_1 = C_2$ and $\tilde{\mathbf{Q}}^\star = \mathbf{Q}^\star$, i.e. $\hat{\mathbf{Q}}^\star = [\mathbf{Q}^\star, \boldsymbol{\xi}^\star]$. We can derive $\mathbf{Q}^\star$ by omitting the last column of $\hat{\mathbf{Q}}^\star$. Therefore, Proposition 1 is proved.

## C  PROOF OF PROPOSITION 2

From the Appendix A, we know the key to proving Proposition 2 is to derive the proximal operator for $F_1, F_2$. Without losing generality, we rewrite the entropic version of Equ.(14) in a more simple and general form, which is detailed as follows:

$$\min_{\hat{\mathbf{Q}} \in \Phi} \epsilon KL(\mathbf{Q}, \exp(-\mathbf{C}/\epsilon)) + \hat{KL}(\mathbf{Q}^\top\mathbf{1}_N, \boldsymbol{\beta}, \boldsymbol{\lambda}) \tag{S34}$$

$$\text{s.t.} \quad \Phi = \{\mathbf{Q} \in \mathbb{R}_+^{N \times K} | \mathbf{Q}\mathbf{1}_K = \boldsymbol{\alpha}\}, \tag{S35}$$

In Equ.(S34), the equality constraint means that $F_1$ is an indicator function, formulated as:

$$F_1(\mathbf{x}, \boldsymbol{\alpha}) = \begin{cases} 0, & \mathbf{x} = \boldsymbol{\alpha} \\ +\infty, & \text{otherwise} \end{cases} \tag{S36}$$

Therefore, we can plug in the above $F_1$ to Equ.(S24), and derive the proximal operator for $F_1$ is $\boldsymbol{\alpha}$. For the weighted $KL$ constraint, the proximal operator is as follows:

$$\text{prox}_{F_2/\epsilon}^{KL}(\mathbf{z}, \boldsymbol{\beta}) = \text{argmin}_{\mathbf{x} \in \mathbb{R}_+^K} \hat{K}L(\mathbf{x}, \boldsymbol{\beta}, \boldsymbol{\lambda}) + \epsilon KL(\mathbf{x}, \mathbf{z}) \tag{S37}$$

$$= \text{argmin}_{\mathbf{x} \in \mathbb{R}_+^K} \sum_i \boldsymbol{\lambda}_i \mathbf{x}_i \log \frac{\mathbf{x}_i}{\boldsymbol{\beta}_i} - \boldsymbol{\lambda}_i \mathbf{x}_i + \boldsymbol{\beta}_i + \epsilon \mathbf{x}_i \log \frac{\mathbf{x}_i}{\mathbf{z}_i} - \epsilon \mathbf{x}_i + \mathbf{z}_i \tag{S38}$$

$$= \text{argmin}_{\mathbf{x} \in \mathbb{R}_+^K} \sum_i \boldsymbol{\lambda}_i \mathbf{x}_i \log \frac{\mathbf{x}_i}{\boldsymbol{\beta}_i} - \boldsymbol{\lambda}_i \mathbf{x}_i + \epsilon \mathbf{x}_i \log \frac{\mathbf{x}_i}{\mathbf{z}_i} - \epsilon \mathbf{x}_i \tag{S39}$$

$$= \text{argmin}_{\mathbf{x} \in \mathbb{R}_+^K} \sum_i (\boldsymbol{\lambda}_i + \epsilon)\mathbf{x}_i \log \mathbf{x}_i - (\boldsymbol{\lambda}_i \log \boldsymbol{\beta}_i + \boldsymbol{\lambda}_i + \epsilon \log \mathbf{z}_i + \epsilon)\mathbf{x}_i \tag{S40}$$

The first line to the second is that we expand the unnormalized $KL$. The second line to the third is that we omit the variable unrelated to $\mathbf{x}$. The third line to the fourth line is that we reorganize the equation. Now, we consider a function as follows:

$$g(x) = ax \log x - bx \tag{S41}$$

Its first derivative is denoted as:

$$g'(x) = a + a \log x - b \tag{S42}$$

Therefore, $\text{argmin}_x g(x) = \exp(\frac{b-a}{a})$. According to this conclusion, we know that:

$$\mathbf{x}_i = \exp(\frac{\boldsymbol{\lambda}_i \log \boldsymbol{\beta}_i + \epsilon \log \mathbf{z}_i}{\boldsymbol{\lambda}_i + \epsilon}) = \boldsymbol{\beta}_i^{\frac{\boldsymbol{\lambda}_i}{\boldsymbol{\lambda}_i + \epsilon}} \mathbf{z}_i^{\frac{\epsilon}{\boldsymbol{\lambda}_i + \epsilon}} \tag{S43}$$

And the vectorized version is as follows::

$$\mathbf{x} = \boldsymbol{\beta}^{\circ \boldsymbol{f}} \mathbf{z}^{\circ(1-\boldsymbol{f})}, \quad \boldsymbol{f} = \frac{\boldsymbol{\lambda}}{\boldsymbol{\lambda} + \epsilon}. \tag{S44}$$

Finally, we plug in the two proximal operators into Alg.2. The iteration of $\mathbf{a}, \mathbf{b}$ is as follows:

$$\mathbf{a} \leftarrow \text{prox}_{F_1/\epsilon}^{KL}(\mathbf{Mb}, \boldsymbol{\alpha})/(\mathbf{Mb}) = \frac{\boldsymbol{\alpha}}{\mathbf{M} \cdot \mathbf{b}} \tag{S45}$$

$$\mathbf{b} \leftarrow \text{prox}_{F_2/\epsilon}^{KL}(\mathbf{M}^\top \mathbf{a}, \boldsymbol{\beta})/(\mathbf{M}^\top \mathbf{a}) = \boldsymbol{\beta}^{\circ \boldsymbol{f}}(\mathbf{M}^\top \mathbf{a})^{\circ(1-\boldsymbol{f})}/(\mathbf{M}^\top \mathbf{a}) = (\frac{\boldsymbol{\beta}}{\mathbf{M}^\top \mathbf{a}})^{\circ \boldsymbol{f}} \tag{S46}$$

Consequently, Proposition 2 is proved.

## D  MORE DISCUSSION ON UNIFORM CONSTRAINT

In deep clustering, the uniform constraint is widely used to avoid degenerate solutions. Among them, KL constraint is a common feature in many clustering baselines, such as SCAN (Van Gansbeke et al., 2020), PICA(Huang et al., 2020), CC(Li et al., 2021), and DivClust(Metaxas et al., 2023). These methods employ an additional entropy regularization loss term on cluster size to prevent degenerate solutions, which is equivalent to the KL constraint. Different from above, Huang et al. (2022) achieves uniform constraints by maximizing the inter-clusters distance between prototypical representations. By contrast, our method incorporates KL constraint in pseudo-label generation, constituting a novel OT problem. As demonstrated in 1, our approach significantly outperforms these baselines.

## E  DATASETS DETAILS

Fig.S6 displays the class distribution of ImgNet-R and several training datasets from iNaturalist18. As evident from the figure, the class distribution in several iNature datasets is highly imbalanced. While ImgNet-R exhibits a lower degree of imbalance, its data distribution still presents a significant challenge for clustering tasks.

## F  MORE IMPLEMENTATION DETAILS

In this section, we illustrate the implementation details of other methods.

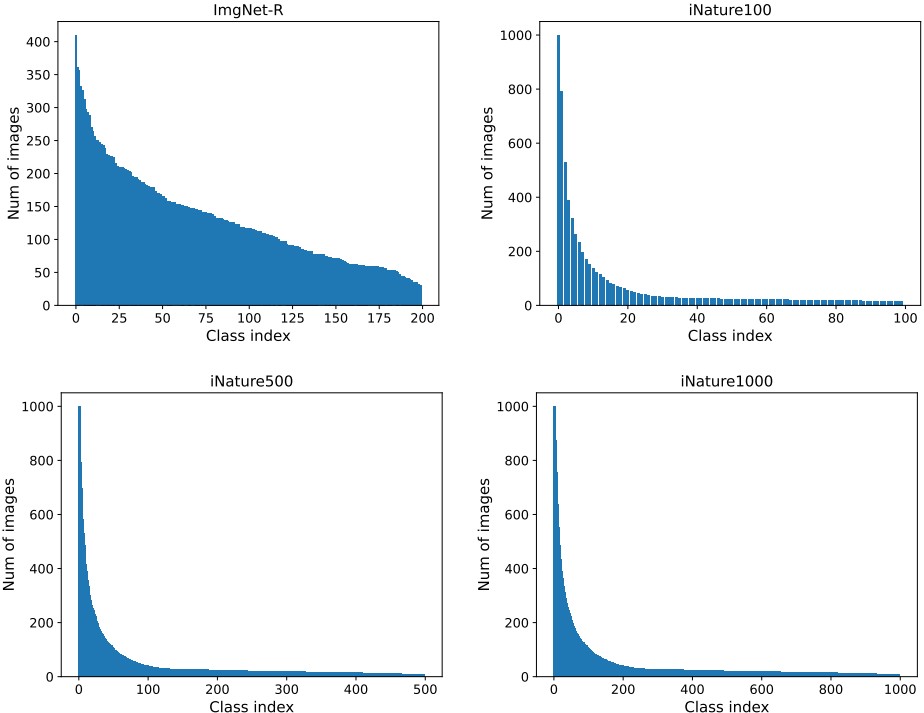

Figure S6: The class distribution of different datasets.

## F.1 DEEP CLUSTERING

In our study, we employed various methods, all utilizing the same datasets and finetuning the last block of ViT-B16. All methods are trained for 50 epochs due to convergence observation.

For our P²OT method, we use a batch size of 512. To stabilize the mini-batch P²OT, We incorporate a memory buffer queue of size 5120 for history prediction after the first epoch to stabilize optimization. During training, we apply two transformations to the same sample and cross-utilize the two pseudo-labels as supervision, thereby encouraging the model to maintain equivariance. We employ 2 clustering heads on the CIFAR100 dataset and 1 clustering head on others. The clustering heads are designed with the sample cluster number as ground truth. Since the sum of sample weights in the pseudo label $\mathbf{Q}$ is $\rho$, we adjust the loss on training sets for the clustering head and model selection as $\mathcal{L}/\rho$.

For IIC, we adopt their method (loss function) from GitHub. They aim at maximizing the mutual information of class assignments between different transformations of the same sample. In all experiments, we use a batch size of 512. We follow their over-clustering strategy, i.e., using two heads, one matching the ground-truth cluster number and the other set at 200 for CIFAR100 and iNature100, 400 for ImgNet-R, and 700 for iNature500. We do not use over-clustering for iNature1000 due to resource and efficiency considerations. Models from the last epoch are selected.

For PICA, we adopt their method (loss function) from GitHub. Their basic idea is to minimize cluster-wise Assignment Statistics Vector(ASV) cosine similarity. In our implementation, the batch size, over-clustering strategy, and model selection are the same as IIC above.

For SCAN, we adopt their code from GitHub. For clustering, their basic idea is to make use of the information of nearest neighbors from pretrained model embeddings. We use a batch size of 512. Following their implementation, we use 10 heads for CIFAR100, iNature100, iNature500, and ImgNet-R. We use 1 head for iNature1000 due to resource and efficiency considerations. While the authors select models based on loss on test sets, we argue that this strategy may not be suitable as test sets are unavailable during training. For the fine-tuning stage of SCAN, they use high-confident samples to obtain pseudo labels. Following their implementation, we use the confidence threshold

of 0.99 on CIFAR. On other datasets, we use the confidence threshold of 0.7 because the model is unable to select confident samples if we use 0.99.

For CC, we adopt their code from GitHub. They perform both instance- and cluster-level contrastive learning. Following their implementation, we use a batch size of 128, and the models of the last epochs are selected.

For DivClust, we adopt their code from GitHub. Their basic idea is to enlarge the diversity of multiple clusterings. Following their implementation, we use a batch size of 256, with 20 clusterings for CIFAR100, ImgNet-R, iNature100, and iNature500. We use 2 clusterings for iNature1000 due to resource and efficiency considerations. The models of the last epoch along with the best heads are selected.

Table S4: SPICE accuracy on different datasets.

| Dataset | CIFAR100 | ImgNet-R | iNature100 | iNature500 | iNature1000 |
| Imbalance ratio | 100 | 13 | 67 | 111 | 111 |
|---|---|---|---|---|---|
| SPICE | 6.74 | 2.58 | 10.29 | 2.15 | 0.74 |

We also tried to implement SPICE from GitHub on imbalanced datasets. Given a pre-trained backbone, they employ a two-stage training process: initially training the clustering head while keeping the backbone fixed, and subsequently training both the backbone and clustering head jointly. For the first stage, they select an equal number of confident samples for each cluster and generate pseudo-labels to train the clustering head. However, on imbalanced datasets, this strategy can easily lead to degraded results as presented in Tab.S4. For example, for the medium and tail classes, the selected confident samples will easily include many samples from head classes, causing clustering head degradation. For the second stage, they utilize neighbor information to generate reliable pseudo-labels for further training. However, the degradation of the clustering head, which occurs as a result of the first stage's limitations, poses a hurdle in the selection of reliable samples for the second stage, thus preventing the training of the second stage. For comparison with SPICE on the balanced training set, please refer to Sec.I.

### F.2 Representation Learning with long-tailed data

In the literature, (Zhou et al., 2022b; Liu et al., 2022; Jiang et al., 2021) aims to learn unbiased representation with long-tailed data. For a fair comparison, we adopt the pre-trained ViT-B16, finetune the last block of the ViT-B16 with their methods by 500 epochs, and perform K-means on representation space to evaluate the clustering ability. However, we exclude reporting results from (Liu et al., 2022; Jiang et al., 2021) for the following reasons: 1) (Jiang et al., 2021) prunes the ResNet, making it challenging to transfer to ViT; 2) (Liu et al., 2022) has not yet open-sourced the code for kernel density estimation, which is crucial for its success.

## G ARI metric

Additionally, we report the results on the Adjusted Rand Index (ARI) metric. Interestingly, on the imbalanced training set (Fig.S5), methods like IIC and SCAN* tend to exhibit a bias toward head classes (Fig.S7) and achieve very high ARI scores, even though they perform inferiorly compared to our method in terms of ACC and NMI. However, as shown in Fig.S6, on the balanced test set, the performance of IIC and SCAN* is relatively poor compared to our method. This inconsistency in performance across different class distributions highlights that ARI can be sensitive to the distribution of data and may not be a suitable metric in imbalance scenarios.

## H More Comparison

**iNature1000 dataset.** As shown in Tab.S7, in the large-scale iNature1000 dataset, our method achieves comparable results with Strong SCAN. Specifically, we achieve improvements of 0.5 in ACC and 0.3 in F1 score, while there is a slight decrease of 0.6 in the NMI metric. It's important to note that pseudo-labeling-based methods face significant challenges in large-scale scenarios, as

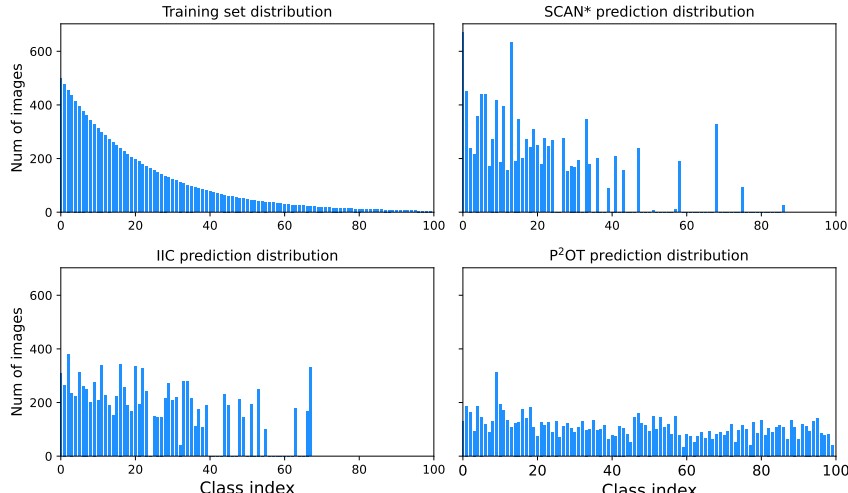

Figure S7: Prediction distribution comparison of IIC, SCAN* and P$^2$OT on CIFAR100.

Table S5: ARI Comparison with SOTA methods on imbalanced training datasets.

| Dataset
Imbalance ratio | CIFAR100
100 | ImgNet-R
13 | iNature100
67 | iNature500
111 | iNature1000
111 |
|---|---|---|---|---|---|
| IIC | $43.3_{\pm 3.9}$ | $13.2_{\pm 0.8}$ | $\mathbf{33.6}_{\pm 6.2}$ | $19.2_{\pm 0.6}$ | $20.2_{\pm 1.5}$ |
| PICA | $24.6_{\pm 0.2}$ | $5.2_{\pm 0.1}$ | $12.6_{\pm 0.5}$ | $7.1_{\pm 0.1}$ | $5.4_{\pm 0.2}$ |
| SCAN | $34.7_{\pm 0.5}$ | $13.5_{\pm 0.4}$ | $18.2_{\pm 0.2}$ | $13.6_{\pm 0.1}$ | $14.3_{\pm 0.2}$ |
| SCAN* | $\mathbf{50.0}_{\pm 2.4}$ | $15.8_{\pm 0.2}$ | $21.0_{\pm 0.4}$ | $\mathbf{27.6}_{\pm 1.3}$ | $\mathbf{28.0}_{\pm 1.6}$ |
| CC | $29.8_{\pm 0.4}$ | $6.0_{\pm 0.3}$ | $13.9_{\pm 0.8}$ | $13.6_{\pm 1.3}$ | $11.9_{\pm 0.6}$ |
| DivClust | $29.0_{\pm 1.1}$ | $7.1_{\pm 0.2}$ | $14.2_{\pm 0.5}$ | $11.0_{\pm 0.5}$ | $10.8_{\pm 0.9}$ |
| P$^2$OT | $36.9_{\pm 1.3}$ | $\mathbf{16.0}_{\pm 0.7}$ | $23.1_{\pm 0.5}$ | $14.3_{\pm 0.3}$ | $12.6_{\pm 0.2}$ |

Table S6: ARI Comparison with SOTA methods on balanced test datasets.

| Dataset
Imbalance ratio | CIFAR100
100 | ImgNet-R
13 | iNature100
67 | iNature500
111 | iNature1000
111 |
|---|---|---|---|---|---|
| IIC | $21.5_{\pm 1.6}$ | $11.6_{\pm 0.9}$ | $16.9_{\pm 1.7}$ | $7.7_{\pm 0.4}$ | $4.4_{\pm 0.1}$ |
| PICA | $19.9_{\pm 0.5}$ | $5.2_{\pm 0.0}$ | $17.6_{\pm 3.4}$ | $6.4_{\pm 0.3}$ | $3.6_{\pm 0.2}$ |
| SCAN | $28.2_{\pm 0.2}$ | $12.0_{\pm 0.5}$ | $25.0_{\pm 1.7}$ | $17.0_{\pm 0.3}$ | $14.0_{\pm 0.3}$ |
| SCAN* | $21.7_{\pm 1.8}$ | $\mathbf{13.9}_{\pm 0.1}$ | $27.7_{\pm 0.6}$ | $11.1_{\pm 0.6}$ | $5.8_{\pm 0.2}$ |
| CC | $19.1_{\pm 0.7}$ | $4.3_{\pm 0.3}$ | $15.1_{\pm 0.4}$ | $6.8_{\pm 0.5}$ | $4.3_{\pm 0.1}$ |
| DivClust | $21.3_{\pm 0.6}$ | $5.5_{\pm 0.5}$ | $19.2_{\pm 1.1}$ | $8.0_{\pm 0.6}$ | $4.5_{\pm 0.3}$ |
| P$^2$OT | $\mathbf{28.8}_{\pm 0.8}$ | $13.4_{\pm 0.8}$ | $\mathbf{29.7}_{\pm 1.7}$ | $\mathbf{18.9}_{\pm 0.4}$ | $\mathbf{14.2}_{\pm 0.7}$ |

generating high-quality pseudo-labels becomes increasingly difficult. Despite this challenge, our method demonstrates competitive performance in such a challenging setting.

**Results on the balanced test sets.** As presented in Tab.S8, we conduct a comprehensive comparison of our method with existing approaches on balanced test sets, validating their ability in an inductive setting. On the relatively small-scale CIFAR100 dataset, our method outperforms the previous state-of-the-art (SCAN) by margins of 1.4, respectively. On the ImgNet-R datasets, our method demonstrates its effectiveness with a notable improvement of 2.2, highlighting its robustness in out-of-distribution scenarios. On the fine-grained iNature datasets, our performance gains are substantial across each subset. Particularly, we observe improvements of 5.2 on iNature100, 3.2 on iNature500, and 1.2 on iNature1000. Note that SPCIE degrades in the imbalance scenario (see Appendix F), and SCAN* performs worse than SCAN in some cases, indicating the ineffectiveness of naive

Table S7: Comparison with SOTA methods on iNature1000 training set.

| Method | iNature1000 | | |
|--------|-----|-----|-----|
| | ACC | NMI | F1 |
| IIC | $7.8_{\pm0.5}$ | $56.9_{\pm0.4}$ | $4.2_{\pm0.2}$ |
| PICA | $12.4_{\pm0.2}$ | $55.0_{\pm0.1}$ | $8.5_{\pm0.1}$ |
| SCAN | $\underline{26.2}_{\pm0.2}$ | $\mathbf{68.0}_{\pm0.0}$ | $\underline{19.6}_{\pm0.1}$ |
| SCAN* | $10.5_{\pm0.2}$ | $63.9_{\pm0.5}$ | $6.4_{\pm0.2}$ |
| CC | $12.1_{\pm0.4}$ | $55.2_{\pm0.4}$ | $9.1_{\pm0.4}$ |
| DivClust | $13.0_{\pm0.2}$ | $56.0_{\pm0.3}$ | $9.5_{\pm0.2}$ |
| $P^2OT$ | $\mathbf{26.7}_{\pm0.5}$ | $\underline{67.4}_{\pm0.4}$ | $\mathbf{19.9}_{\pm0.4}$ |

Table S8: ACC Comparison with SOTA methods on balanced test sets. SCAN* (Van Gansbeke et al., 2020) denotes performing another self-labeling training stage based on SCAN.

| Dataset | CIFAR100 | ImgNet-R | iNature100 | iNature500 | iNature1000 |
|---------|----------|----------|------------|------------|-------------|
| Imbalance ratio | 100 | 13 | 67 | 111 | 111 |
| IIC | $29.4_{\pm3.0}$ | $21.4_{\pm1.7}$ | $35.1_{\pm2.7}$ | $19.8_{\pm0.7}$ | $13.3_{\pm1.0}$ |
| PICA | $30.3_{\pm0.4}$ | $16.6_{\pm0.1}$ | $43.5_{\pm3.3}$ | $31.4_{\pm0.3}$ | $28.3_{\pm0.2}$ |
| SCAN | $\underline{37.5}_{\pm0.3}$ | $23.8_{\pm0.8}$ | $44.0_{\pm0.9}$ | $\underline{39.0}_{\pm0.1}$ | $\underline{36.5}_{\pm0.3}$ |
| SCAN* | $30.5_{\pm0.6}$ | $\underline{25.2}_{\pm0.1}$ | $\underline{44.8}_{\pm0.7}$ | $24.3_{\pm0.6}$ | $14.5_{\pm0.2}$ |
| CC | $29.1_{\pm1.3}$ | $14.7_{\pm0.2}$ | $38.2_{\pm1.4}$ | $29.7_{\pm1.1}$ | $26.6_{\pm0.5}$ |
| DivClust | $31.8_{\pm0.6}$ | $16.8_{\pm0.3}$ | $42.4_{\pm1.2}$ | $31.6_{\pm0.5}$ | $26.9_{\pm0.1}$ |
| $P^2OT$ | $\mathbf{38.9}_{\pm1.1}$ | $\mathbf{27.5}_{\pm1.2}$ | $\mathbf{50.0}_{\pm2.1}$ | $\mathbf{42.2}_{\pm1.1}$ | $\mathbf{37.7}_{\pm0.2}$ |

Table S9: NMI Comparison with SOTA methods on balanced test sets.

| Dataset | CIFAR100 | ImgNet-R | iNature100 | iNature500 | iNature1000 |
|---------|----------|----------|------------|------------|-------------|
| Imbalance ratio | 100 | 13 | 67 | 111 | 111 |
| IIC | $56.0_{\pm1.6}$ | $53.9_{\pm1.0}$ | $75.4_{\pm1.4}$ | $73.9_{\pm0.4}$ | $71.2_{\pm0.6}$ |
| PICA | $54.5_{\pm0.5}$ | $51.9_{\pm0.1}$ | $78.2_{\pm1.5}$ | $79.7_{\pm0.2}$ | $80.0_{\pm0.2}$ |
| SCAN | $\underline{63.0}_{\pm0.1}$ | $57.0_{\pm0.5}$ | $80.8_{\pm0.5}$ | $\underline{83.4}_{\pm0.1}$ | $\underline{84.2}_{\pm0.1}$ |
| SCAN* | $57.6_{\pm2.0}$ | $\underline{57.1}_{\pm0.3}$ | $\underline{81.9}_{\pm0.6}$ | $77.0_{\pm0.6}$ | $72.8_{\pm0.3}$ |
| CC | $54.2_{\pm0.6}$ | $48.5_{\pm0.3}$ | $76.3_{\pm0.3}$ | $78.3_{\pm0.7}$ | $78.7_{\pm0.2}$ |
| DivClust | $58.1_{\pm0.3}$ | $50.4_{\pm0.3}$ | $78.7_{\pm0.6}$ | $79.2_{\pm0.5}$ | $78.8_{\pm0.2}$ |
| $P^2OT$ | $\mathbf{63.1}_{\pm0.8}$ | $\mathbf{58.0}_{\pm0.7}$ | $\mathbf{83.0}_{\pm0.9}$ | $\mathbf{84.3}_{\pm0.1}$ | $\mathbf{84.4}_{\pm0.2}$ |

Table S10: F1 Comparison with SOTA methods on balanced test sets.

| Dataset | CIFAR100 | ImgNet-R | iNature100 | iNature500 | iNature1000 |
|---------|----------|----------|------------|------------|-------------|
| Imbalance ratio | 100 | 13 | 67 | 111 | 111 |
| IIC | $21.0_{\pm3.6}$ | $17.7_{\pm1.9}$ | $25.7_{\pm2.8}$ | $10.6_{\pm0.4}$ | $5.0_{\pm0.7}$ |
| PICA | $28.2_{\pm0.6}$ | $16.6_{\pm0.2}$ | $\underline{38.7}_{\pm3.5}$ | $28.3_{\pm0.2}$ | $26.0_{\pm0.1}$ |
| SCAN | $\underline{34.9}_{\pm0.4}$ | $\underline{23.8}_{\pm0.9}$ | $36.5_{\pm1.1}$ | $\underline{33.3}_{\pm0.3}$ | $\underline{31.8}_{\pm0.4}$ |
| SCAN* | $19.8_{\pm0.8}$ | $23.5_{\pm0.2}$ | $36.8_{\pm1.3}$ | $14.4_{\pm0.4}$ | $5.8_{\pm0.2}$ |
| CC | $25.6_{\pm1.3}$ | $14.2_{\pm0.4}$ | $32.4_{\pm0.6}$ | $26.5_{\pm1.4}$ | $23.9_{\pm0.8}$ |
| DivClust | $28.9_{\pm0.6}$ | $16.4_{\pm0.3}$ | $36.1_{\pm1.4}$ | $27.8_{\pm0.6}$ | $24.3_{\pm0.2}$ |
| $P^2OT$ | $\mathbf{35.9}_{\pm0.8}$ | $\mathbf{28.1}_{\pm1.3}$ | $\mathbf{44.2}_{\pm2.1}$ | $\mathbf{37.2}_{\pm1.4}$ | $\mathbf{33.0}_{\pm0.2}$ |

self-labeling training. These strong results demonstrate the superiority of our method in fine-grained and large-scale scenarios.

The NMI and F1 result comparisons of our method with existing approaches are presented in Tab.S9 and Tab.S10. For NMI, our method achieves comparable results with previous state-of-the-art (SCAN) on the CIFAR100 and iNature1000 datasets. On ImgNet-R, iNature100, and iNature500, our method outperforms state-of-the-art by $\sim 1\%$. For F1, our method surpasses the previous method by a large margin, demonstrating the effectiveness of our method in handling imbalance learning.

Table S11: Comparision with SPICE on balanced training set

| Method | Backbone | Fine Tune | Confidence Ratio | Time Cost(h) | CIFAR100 | | |
|--------|----------|-----------|------------------|--------------|------|------|------|
| | | | | | ACC | NMI | ARI |
| SPICE$_s$ | ViT-B16 | False | - | 3.6 | 31.9 | 57.7 | 22.5 |
| SPICE$_s$ | ViT-B16 | True | - | 6.7 | 60.9 | 69.2 | 46.3 |
| SPICE | ViT-B16 | False | >0 | 3.6 | Fail | Fail | Fail |
| SPICE | ViT-B16 | True | >0.6 | 6.7 | Fail | Fail | Fail |
| SPICE | ViT-B16 | True | 0.5 | 21.6 | **68.0** | **77.7** | **54.8** |
| Ours | ViT-B16 | False | - | 2.9 | 61.9 | 72.1 | 48.8 |

## I COMPARISON AND ANALYSIS ON BALANCED TRAINING DATASET

To compare and analyze our method with baselines on balanced datasets, we conduct experiments using the balanced CIFAR100 dataset as demonstrated in Tab.S11. Overall, we have achieved superior performance on imbalanced datasets, and on balanced datasets, we have achieved satisfactory results. However, in balanced settings, we believe that it is a bit unfair to compare our proposed method to methods developed in balanced scenarios, which usually take advantage of the uniform prior.

Note that, as imagenet-r and iNaturelist2018 subsets are inherently imbalanced, we do not conduct experiments on them.

For all methods, we employ the ImageNet-1K pre-trained ViT-B16 backbone. Fine tune refers to the first training stage of SPICE, which trains the model on the target dataset using self-supervised learning, like MoCo. SPICE$_s$ corresponds to the second stage of training, which fixes the backbone and learns the clustering head by selecting top-K samples for each class, heavily relying on the uniform distribution assumption. SPICE refers to the final joint training stage, which selects an equal number of reliable samples for each class based on confidence ratio and applies FixMatch-like training. In SPICE, the confidence ratio varies across different datasets. Time cost denotes the accumulated time cost in hours on A100 GPUs.

The observed results in the table reveal several key findings: 1) Without self-supervised training on the target dataset, SPICE$_s$ yields inferior results; 2) SPICE tends to struggle without self-supervised training and demonstrates sensitivity to hyperparameters; 3) SPICE incurs a prolonged training time; 4) Our method achieves comparable results with SPICE$_s$ and inferior results than SPICE.

On imbalanced datasets, as demonstrated in Tab.S4, SPICE encounters difficulties in clustering tail classes due to the noisy top-K selection, resulting in a lack of reliable samples for these classes. In contrast, our method attains outstanding performance on imbalanced datasets without the need for further hyperparameter tuning.

In summary, the strength of SPICE lies in its ability to leverage the balanced prior and achieve superior results in balanced scenarios. However, SPICE exhibits weaknesses: 1) It necessitates multi-stage training, taking 21.6 hours and 7.45 times our training cost; 2) It requires meticulous tuning of multiple hyperparameters for different scenarios and may fail without suitable hyperparameters, while our method circumvents the need for tedious hyperparameter tuning; 3) It achieves unsatisfactory results on imbalanced datasets due to its strong balanced assumption.

In contrast, the advantage of our method lies in its robustness to dataset distribution, delivering satisfactory results on balanced datasets and superior performance on imbalanced datasets.

## J FURTHER ANALYSIS OF POT, UOT AND SLA

In this section, we provide a comprehensive explanation of the formulations of POT, UOT, and SLA (Tai et al., 2021), followed by an analysis of their limitations in the context of deep imbalanced scenarios. POT replaces the $KL$ constraint with an equality constraint, and its formulation is as

---

**Algorithm 3:** Generalized Scaling Algorithm for P$^2$OT

---

**Input:** Cost matrix $-\log \mathbf{P}$, $\epsilon$, $\lambda$, $\rho$, $N$, $K$
$\mathbf{C} \leftarrow -\log \mathbf{P}$
$\boldsymbol{\beta} \leftarrow \frac{\rho}{K}\mathbf{1}_K^\top, \quad \boldsymbol{\alpha} \leftarrow \frac{1}{N}\mathbf{1}_N,$
$\mathbf{b} \leftarrow \mathbf{1}_K, \quad s \leftarrow 1, \quad \mathbf{M} \leftarrow \exp(-\mathbf{C}/\epsilon), \quad \boldsymbol{f} \leftarrow \frac{\lambda}{\lambda+\epsilon}$
**while** $b$ *not converge* **do**
$\quad \mathbf{a} \leftarrow min(\frac{\boldsymbol{\alpha}}{s\mathbf{Mb}}, 1)$
$\quad \mathbf{b} \leftarrow (\frac{\boldsymbol{\beta}}{s\mathbf{M}^\top \mathbf{a}})^{\boldsymbol{f}}$
$\quad s \leftarrow \frac{\rho}{\mathbf{a}^\top \mathbf{Mb}}$
**end**
**return** diag($\mathbf{a}$)$\mathbf{M}$diag($\mathbf{b}$);

---

follows:

$$\min_{\mathbf{Q}\in\Pi}\langle\mathbf{Q}, -\log\mathbf{P}\rangle_F \tag{S47}$$

$$\text{s.t.} \quad \Pi = \{\mathbf{Q}\in\mathbb{R}_+^{N\times K}|\mathbf{Q}\mathbf{1}_K \leq \frac{1}{N}\mathbf{1}_N, \mathbf{Q}^\top\mathbf{1}_N = \frac{\rho}{K}\mathbf{1}_K, \mathbf{1}_N^\top\mathbf{Q}\mathbf{1}_K = \rho\} \tag{S48}$$

This formulation overlooks the class imbalance distribution, thus making it hard to generate accurate pseudo-labels in imbalance scenarios.

On the other hand, UOT eliminates the progressive $\rho$, and its corresponding formulation is denoted as follows:

$$\min_{\mathbf{Q}\in\Pi}\langle\mathbf{Q}, -\log\mathbf{P}\rangle_F + \lambda KL(\mathbf{Q}^\top\mathbf{1}_N, \frac{1}{K}\mathbf{1}_K) \tag{S49}$$

$$\text{s.t.} \quad \Pi = \{\mathbf{Q}\in\mathbb{R}_+^{N\times K}|\mathbf{Q}\mathbf{1}_K = \frac{1}{N}\mathbf{1}_N\} \tag{S50}$$

UOT has the capability to generate imbalanced pseudo-labels, but it lacks the ability to effectively reduce the impact of noisy samples, which can adversely affect the model's learning process.

SLA (Tai et al., 2021), represented by Equ.(S51), relaxes the equality constraint using an upper bound $b$. However, during the early training stages when $\rho$ is smaller than the value in $b\mathbf{1}_K$, SLA tends to assign all the mass to a single cluster, leading to a degenerate representation.

$$\min_{\mathbf{Q}\in\Pi}\langle\mathbf{Q}, -\log\mathbf{P}\rangle_F \tag{S51}$$

$$\text{s.t.} \quad \Pi = \{\mathbf{Q}\in\mathbb{R}_+^{N\times K}|\mathbf{Q}\mathbf{1}_K \leq \frac{1}{N}\mathbf{1}_N, \mathbf{Q}^\top\mathbf{1}_N \leq b\mathbf{1}_K, \mathbf{1}_N^\top\mathbf{Q}\mathbf{1}_K = \rho\} \tag{S52}$$

## K    GENERALIZED SCALING ALGORITHM FOR P$^2$OT

We provide the pseudo-code for the Generalized Scaling Algorithm (GSA) proposed by (Chizat et al., 2018) in Alg.3.

