# OpenReview forum: "P$^2$OT: Progressive Partial Optimal Transport for Deep Imbalanced Clustering"
_ICLR.cc/2024/Conference — ICLR 2024 poster_

### Official Review · Reviewer_qtAk · 2023-10-31

**Soundness:** 3 good
**Presentation:** 3 good
**Contribution:** 3 good
**Rating:** 6
**Confidence:** 4

**Summary:**

This paper studied a more general clustering problem, deep imbalanced clustering. From the perspective of pseudo label generation, the authors propose a progressive partial optimal transport method to combat the imbalanced challenges in data. The technical way to incorporate the imbalanced distribution into the optimal-transport framework and align with the classical solver is appealing, and a range of experiments demonstrate the performance of the proposed method.

**Strengths:**

1) The technical point is good, which leverages a virtual cluster to incorporate the spirit of sample selection in optimal transport is novel. The authors applied a mass increasing process in the constraint of unbalanced OT to progressively leverage more confident samples for representation learning and avoids the degeneration due to the skew distribution.

2) The reformulation with entropy regularization and the weighted the KL divergence makes the classical Sinkhorn-knop algorithm appliable and stably optimize the target towards the progressive imbalance constraint. The authors carefully deal with the reformulation to make the algorithm can sufficiently incorporate the desire for sample selection.

3) The authors conduct a range of experiments on the representative datasets like small dataset CIFAR100-LT, the mid-scale ImageNet-R, and large-scale iNaturalist. The experimental results and the visualization consistently support the author's claim, and the ablation study provided the insight on how the components work and how useful they are.

**Weaknesses:**

Although the proposed method is overall novel, there are still some concerns that should be considered.
1) The technical choice about Eq.(13) is unique. The other choice like introducing a hard equality constraint about the mass of the virtual cluster and applying the lagrange multiplier can be also possible. The main concern here is that the authors introduce two hyperparameters for weighted KL: one is for the target clusters, i.e., \lambda, and the other is the large value for the virtual cluster.  How is the performance of directly constraining the equality about 1-\rou with lagrange multiplier compared with Eq.(13).

2) It is not clear why the authors have not compared with the clustering with self-labeling by Asano from the perspective of OT. For SPICE, the performance reported in the appendix is also not convincing. How is the performance comparison of SPICE and P^2OT on the balanced datasets. It will be provide more comprehensive comparison here on both imbalanced datasets and balanced datasets to show their pros and cons.

3) In the perspective of representation learning, some related works should be included with the proper discussion, e.g., about the self-supervised long-tailed learning, like SDCLR [1], BCL [2] and re-weighted regularization, as they also target to representation learning of imbalanced data without label information. Some proper comparison will be better.

[1] Jiang, Z., Chen, T., Mortazavi, B.J. and Wang, Z., 2021, July. Self-damaging contrastive learning. In International Conference on Machine Learning (pp. 4927-4939). PMLR.
[2] Zhou, Z., Yao, J., Wang, Y.F., Han, B. and Zhang, Y., 2022, June. Contrastive learning with boosted memorization. In International Conference on Machine Learning (pp. 27367-27377). PMLR.
[3] Liu, H., HaoChen, J.Z., Gaidon, A. and Ma, T., 2021. Self-supervised learning is more robust to dataset imbalance. In International Conference on Learning Representation, 2022.

**Questions:**

1) The performance of P^2OT and SPICE on the balanced datasets.

2) How is the performance comparison between the proposed deep imbalanced clustering and some self-supervised long-tailed learning methods in terms of the representation quality?

---

> ### Author Response · Authors · 2023-11-19
> **Response to Reviewer qtAk**
>
> We appreciate the diligent assessment by reviewer qtAk, as well as the recognition of our technical contributions and our proposed methods. We hope the concerns raised could be addressed by the following response.
>
> ## The technical choice about Eq.(13) is unique.
>
> It is possible to apply the Lagrange multiplier to impose a hard equality constraint. But resulting OT problem can not be easily solved by fast scaling algorithm, making it less practical for large-scale application problems. By contrast, we introduce a virtual cluster and weighted KL constraint to transform Eq.5-6 into a form similar to Eq.1, which can be solved by the fast scaling algorithm.
>
> Regarding the additional hyperparameter, we assign $\lambda_{:K}$  as 1 for all datasets without any tuning. And for $\lambda_{K+1}$, we believe that setting it to a sufficiently large value is adequate. In practice, directly assigning the $\frac{\lambda_{K+1}}{\lambda_{K+1}+\epsilon}$ as 1 in Proposition 2 without any further adjustment proves effective.
>
> ## Comparing with OT
> | Formulation | CIFAR100 | ImgNet-R | iNature500 |
> |:-----------:|:--------:|:--------:|:----------:|
> |             |    ACC   |    ACC   |     ACC    |
> |      OT     |   28.2   |   18.2   |    24.1    |
> |   P$^2$OT   |   38.2   |   25.9   |    32.2    |
>
> We thank the reviewer qrAk for pointing out this problem. We have added the results of OT[1] to Table 2, and our method shows considerable improvement over naive OT [1].
>
> As for the comparison of SPICE on balance datasets, please refer to General Response.
>
> [1] Yuki M. Asano, Christian Rupprecht, and Andrea Vedaldi. Self-labelling via simultaneous clustering and representation learning. In International Conference on Learning Representations (ICLR), 2020.
>
> ## Comparison with representation learning.
> Please refer to General Response.

---

> > ### Comment · Reviewer_qtAk · 2023-11-23
> >
> > Thanks for the response. After reading the rebuttal, my major concerns have been addressed and will maintain the score. I hope the authors can take the advices to substantially refine the submission.
> >
> > Best,
> >
> > The reviewer qtAK.

---

> > > ### Author Response · Authors · 2023-11-23
> > > **Response to Reviewer qtAk**
> > >
> > > We sincerely appreciate reviewer suggestions and replies. We have incorporated the outcomes of representation learning into the appendix and will add discussion about imbalanced representation learning directly into the main body of the text.

---

### Official Review · Reviewer_xbkw · 2023-10-31

**Soundness:** 3 good
**Presentation:** 3 good
**Contribution:** 3 good
**Rating:** 6
**Confidence:** 3

**Summary:**

This paper addresses the challenge of deep clustering under imbalanced data. The paper proposes a novel pseudo-labeling-based learning framework. This framework formulates pseudo-label generation as a progressive partial optimal transport problem, allowing it to generate imbalance-aware pseudo-labels and learn from high-confidence samples. The approach transforms the problem into an unbalanced optimal transport problem with augmented constraints, making it efficiently solvable. Experimental results on various datasets, including long-tailed CIFAR100, ImageNet-R, and iNaturalist2018 subsets, demonstrate the effectiveness of the method.

**Strengths:**

This paper studies an interesting problem, deep clustering under imbalanced data. It proposed a progressive partial optimal transport algorithm to address this problem, and extensive experiments have been conducted to evaluate its effectiveness.

**Weaknesses:**

1. Great computational cost. In Figure 5, it takes 1 second to estimate the pseudo-labels when K=40. It is impossible in practice with a larger number of clusters, eg, imagenet, and mini-batch training.
2. Missing comparisons on balanced datasets. The true data distribution is unknown in real-world applications. This paper only investigates the settings of imbalanced data or tests on balanced data. It is not well aligned with most literature of deep clustering.
3. This paper does not learn representations, which may be confusing and needs to be clarified. In addition, this paper uses a large pre-trained model and lacks a simple baseline with representation learning models. For example, BYOL or DNIO pre-trained on ImageNet can be used to extract the representations for subsequent K-means clustering. In such settings, we do not need to determine whether the data is imbalanced, as each sample belongs to a single class or a large number of clusters can be pre-defined.
4. A good evaluation metric should be important for imbalanced deep clustering. Under the imbalanced setting, no samples may be assigned to tail classes during Hungarian matching. As we can see in Figure S7, the predictions of the proposed method are more uniform than baselines. There are more samples assigned to tail classes, though this is not true in training data. It confirms that a uniform clustering result is beneficial for evaluation. I suggest that kNN evaluation in representation learning can be adopted for imbalanced clustering, without the need of Hungarian matching. Due to the uniform constraints, more discussion should be paid to Huang et al., 2022.

Although I have raised many questions, I think this paper is interesting, and I will keep/increase my score if the authors have addressed my concerns.

**Questions:**

1. Change 'confidence sample selection' to 'confident sample selection'
2. Which dataset is used for DINO pretraining? Is the backbone fixed during training?
3. It is unclear about the use of historical predictions.
4. What is visualized in Figure 2? We usually visualize the features before the classifier instead of class predictions. If the results are consistent for the features, we can conclude: more distinct clusters.

---

> ### Author Response · Authors · 2023-11-19
> **Response to Reviewer xbkw**
>
> We appreciate the thorough evaluation by reviewer xbkw and express our sincere gratitude for xbkw's interest in the problem and the method we proposed. We hope the following response could address the concerns raised.
>
> ## Great computation cost
>
> The unit of the horizontal axis is K (kilo), and the analysis is conducted on iNature1000. Thus, when there are 40000 points and 1000 clusters, it takes only 1 second, which is practical for mini-batch training in most cases.
>
> ## Comparisons on Balanced Dataset
>
> Please refer to the general response.
>
> ## This paper does not learn representations.
> We learn the representation. Specifically, we employ an ImageNet-1K unsupervised pre-trained ViT-B16 model and fine-tune the last block, which strikes a balance between complexity and flexibility. We have updated Appendix F to make it clear.
>
> We thank the reviewer's suggestion. We note that K-means clustering is also affected by data imbalance, leading to inferior results and we provide detailed experimental analysis in the second General Response.
>
> ## Evaluation Metric
>
> We appreciate the importance of a good evaluation and recognize that it is an open question in imbalanced deep clustering. However, we disagree that a uniform clustering result is beneficial for evaluation. As evident from the results below, the naive OT, which enforces equality uniform distribution constraints, yields inferior results.
>
> | Formulation | CIFAR100 | ImgNet-R | iNature500 |
> |:-----------:|:--------:|:--------:|:----------:|
> |             |    ACC   |    ACC   |     ACC    |
> |      OT     |   28.2   |   18.2   |    24.1    |
> |   P$^2$OT   |   38.2   |   25.9   |    32.2    |
>
>
> In addition, regarding Hungarian matching, it is true that on imbalanced training sets, tail classes may be less important during the matching process. However, when evaluating on balanced test sets (Tables S8-S10), all classes are equally important during Hungarian matching. Therefore, we can mitigate the bias of Hungarian matching by evaluating on balanced test sets, where we still observe significant improvements.
>
>
> We appreciate reviewer xbKw's suggestion, but we believe that kNN is not a suitable evaluation metric for imbalanced clustering due to the following reasons:
>
> 1. kNN evaluation requires label information from the training set, which is not available in deep clustering.
> 2. kNN is usually an evaluation of representation, while deep clustering learns not only representation but also clustering head.
>
> In conclusion, we suggest evaluating on balanced test set to alleviate the bias of Hungarian matching and evaluate the generalization of different methods.
>
> Huang et al. achieve uniform constraints by maximizing the inter-cluster distance between prototypical representations to avoid degenerate solutions. Unlike [1], we achieve that by utilizing the KL constraint in the pseudo-label generation process. We have updated Appendix D to discuss more on [1] with respect to uniform constraints.
>
>
> [1] Zhizhong Huang, Jie Chen, Junping Zhang, and Hongming Shan. Learning representation for
> clustering via prototype scattering and positive sampling. IEEE Transactions on Pattern Analysis and Machine Intelligence, 2022.
>
>
> ## Questions
> 1. We have corrected the typo.
> 2. The DINO is pre-trained on ImageNet-1K, therefore we have not adopted the ImageNet-1K dataset as a training set. In the training, we only fine-tuned the last transformer block of ViT-B16.
> 3. Please refer to **General Response Part 2**.
> 4. Figure 2 is the visualization of the feature before the cluster head. We have updated the manuscript to make it clear.

---

> > ### Comment · Reviewer_xbkw · 2023-11-21
> > **Response to Authors**
> >
> > Thanks for your efforts in rebuttal. I have read the rebuttal and other reviews. There are no further comments.

---

### Official Review · Reviewer_H8zS · 2023-11-01

**Soundness:** 2 fair
**Presentation:** 2 fair
**Contribution:** 3 good
**Rating:** 6
**Confidence:** 4

**Summary:**

This paper aims to address deep clustering in an imbalanced scenario. In particular, the authors resort to partial optimal transport to gradually select imbalance-aware high-confidence samples based on pseudo-labels. The selected high-confidence samples are then considered as ground truth labelled data for supervised training. The proposed method has been evaluated in human-curated datasets and achieves superior results over baselines.

**Strengths:**

1.	The proposed algorithm is overall reasonable.
2.	Sufficient empirical results are conducted.
3.	The performance over SOTA clustering baselines are impressive.

**Weaknesses:**

1. Some claims in this paper are confusing and need improvement. For instance, (i) "the KL divergence-based uniform distribution constraint empowers our method to avoid degenerate solutions and generate imbalanced pseudo-labels", "demonstrating that KL constraint enables our P2OT to generate imbalanced pseudo labels." Why can the KL constraint generate imbalanced pseudo-labels, given that it is defined for a uniform distribution?
2. I don't think Eq. 5-Eq. 6 are necessary. You can consider introducing Eq. 8 directly after Eq. 3 by introducing a virtual cluster. If I am wrong, please point it out.
3. The technical contributions are overclaimed. It appears that the most important aspect of the proposed method lies in the KL constraints. If these KL constraints are added to other clustering baselines, the superiority of the proposed method will be marginal. Additionally, gradually increasing the number of high-confidence samples has been widely adopted in other clustering papers such as SPCIE. It is true that the proposed method can avoid manual selection, but it still introduces an additional hyperparameter in Eq. 7, i.e., $\rho_0$. More importantly, it does not demonstrate the superiority over the baselines."

**Questions:**

1. A very popular real imbalanced dataset is REUTERS-10K[1], which is more challenging than the constructed dataset. How does the proposed method perform on REUTERS-10K?
2. In Eq. 3, the KL constraint is defined for the uniform distribution. Why it can be called an unbalanced OT problem?
3. The proposed method is defined for the whole dataset. How can it be implemented for a mini-batch scenario? In the mini-batch scenario, samples from minority clusters may not exist.
4. Figure 2 shows that the embedding of P2OT is well-separated compared to that of other baselines. Can the authors explain why the embedding is much better than others, given that it is only slightly higher than SCAN in terms of NMI?
5. Figure 4 shows that the performance degenerates after 10 epochs. Can you explain the reason behind this?

[1] Xie, Junyuan, Ross Girshick, and Ali Farhadi. "Unsupervised deep embedding for clustering analysis." International conference on machine learning. PMLR, 2016

---

> ### Author Response · Authors · 2023-11-19
> **Response to Reviewer H8zS**
>
> We appreciate the valuable feedback and thoughtful comments from reviewer H8zS. We have diligently analyzed the suggestions and concerns raised, and we hope the following response could address them. However, we respectfully disagree with the assessment that "The technical contributions are overclaimed.".
>
> ## Some claims ... are confusing ...
>
> We appreciate that reviewer H8zS points out the confusing statements. We clarify that our method with $KL$ constraint can generate imbalanced pseudo-labels instead of KL can generate imbalanced pseudo-labels. It's worth noting that while the KL constraint is defined for a uniform distribution, it serves as a **relaxed constraint** compared to the typical equality constraint used in [1]. Consequently, the optimal Q in our P$^2$OT should take into account both the inherent imbalanced class distribution in P and a prior uniform distribution. This enables our method to:
> 1. generate pseudo labels reflecting imbalanced characteristics akin to P
> 2. avoid degenerate solutions where all samples are assigned to a single cluster.
>
> For clarity, we have updated the confusion statement in the paper as follows:
> 1. The sentence "the KL divergence-based uniform distribution constraint empowers our method to avoid degenerate solutions and generate imbalanced pseudo-labels" should be changed to: "Notably, the $KL$ divergence-based uniform distribution constraint empowers our method to avoid degenerate solutions. And our approach with $KL$, which represents a relaxed constraint compared to an equality constraint, facilitates the generation of imbalanced pseudo-labels."
> 2. The sentence "demonstrating that KL constraint enables our P$^2$OT to generate imbalanced pseudo labels." should be changed to: "demonstrating that our P2OT with $KL$ constraint can generate imbalanced pseudo labels."
>
>
> [1] Yuki M. Asano, Christian Rupprecht, and Andrea Vedaldi. Self-labeling via simultaneous clustering and representation learning. In International Conference on Learning Representations (ICLR), 2020.
>
> ## Eq.5 - Eq.6 are not necessary.
>
> We respectfully disagree with Reviewer H8zS's assessment of this weakness for the following reasons:
>
> 1. Eq.5-6 formulates the confident sample selection and imbalanced pseudo-label generation as a novel partial optimal transport problem, which is the central component of our method.
>
> 2. Virtual clusters are commonly employed in addressing partial optimal transport (POT) problems. However, Eq.3-4 does not constitute a POT problem. Therefore, introducing Eq.8 directly after Eq.3-4 through the introduction of a virtual cluster makes it challenging to interpret and understand.
>
> In conclusion, we believe that Eq.5-6 is necessary.
>
> [1] Luis A Caffarelli and Robert J McCann. Free boundaries in optimal transport and monge-ampere obstacle problems. Annals of Mathematics, pp. 673–730, 2010.
>
> [2] Laetitia Chapel, Mokhtar Z Alaya, and Gilles Gasso. Partial optimal transport with applications on positive-unlabeled learning. Advances in Neural Information Processing Systems, 33:2903–2913, 2020.

---

> > ### Author Response · Authors · 2023-11-19
> > **Response to Reviewer H8zS**
> >
> > ## The technical contributions are overclaimed.
> > We respond to this weakness sentence by sentence.
> >
> > **It appears that the most important aspect of the proposed method lies in the KL constraints.**
> >
> > Our main contribution lies in our novel P$^2$OT formulation and the efficient solver we developed instead of the solely KL constraint. Our P$^2$OT formulation enables us to select confident samples and generate imbalance-aware pseudo labels by solving a novel optimal transport problem. The efficacy of these components is confirmed by the results presented in Table 2, and the efficiency of our solver is demonstrated in Figure 5.
> >
> > **If these KL constraints are added to other clustering baselines, the superiority of the proposed method will be marginal.**
> >
> > We must emphasize that the KL constraint is already a common feature in many clustering baselines, such as SCAN[1], PICA[2], CC[3], and DivClust[4]. These methods employ an additional entropy regularization loss term on cluster size to prevent degenerate solutions, which is equivalent to the KL constraint. By contrast, our method incorporates KL constraint in pseudo-label generation, constituting a novel OT problem. As demonstrated in Table 1, our approach significantly outperforms these baselines.
> >
> > **...gradually increasing the number of high-confidence samples has been widely adopted...**
> >
> > We posit that the concept of gradually selecting high-confidence samples is intuitive and has been widely employed in semi-supervised learning [5,6,7] and curriculum learning [8,9]. However, the novelty of our approach lies in the method used to achieve it.
> >
> > In general, SPICE relies on manual sample selection through at least two hyper-parameters, necessitating additional training stages. In contrast, our method seamlessly integrates selection and pseudo-labeling within a unified optimal transport framework, requiring only one hyper-parameter and one training stage, which enhances its elegance.
> >
> > Specifically, SPICE, which adopts the concept of gradually selecting high-confidence samples, depends on a pre-trained model with high confidence in the target dataset. This reliance unavoidably introduces multi-stage training. As noted in the General Response, SPICE exhibits sensitivity to hyperparameters. Poorly chosen hyper-parameter thresholds can lead to performance degradation and, in some cases, failure to select any confident samples. In contrast, our hyper-parameter $\rho_0$ proves to be more robust than SPICE, as demonstrated in Tab.3 of our manuscript, consistently delivering strong performance without failures.
> >
> >
> > **More importantly, it does not demonstrate superiority over the baselines.**
> >
> > We note that our method shows superiority over baselines.
> > 1. Table 1 shows our method outperforms previous SOTA by a sizeable margin.
> > 2. Table 2 shows that our method significantly outperforms UOT, demonstrating the effectiveness of selection.
> > 3. Table 3 shows that increasing $\rho$ from 0.1 to 1 leads to a 0.8% improvement in CIFAR100, and 2.3% improvement in iNature500.
> >
> > Therefore, we believe that our formulation is novel and shows great superiority over the baselines.
> >
> >
> > [1] Wouter Van Gansbeke, Simon Vandenhende, Stamatios Georgoulis, Marc Proesmans, and Luc Van Gool. Scan: Learning to classify images without labels. In Proceedings of the European Conference on Computer Vision, 2020.
> >
> > [2] Jiabo Huang, Shaogang Gong, and Xiatian Zhu. Deep semantic clustering by partition confidence maximization. In Proceedings of IEEE Conference on Computer Vision and Pattern Recognition
> > (CVPR), 2020.
> >
> > [3] Yunfan Li, Peng Hu, Zitao Liu, Dezhong Peng, Joey Tianyi Zhou, and Xi Peng. Contrastive clustering. In Proceedings of the AAAI Conference on Artificial Intelligence, volume 35, pp. 8547–8555, 2021.
> >
> > [4] Ioannis Maniadis Metaxas, Georgios Tzimiropoulos, and Ioannis Patras. Divclust: Controlling diversity in deep clustering. In Proceedings of the IEEE/CVF Conference on Computer Vision and Pattern Recognition, pp. 3418–3428, 2023.
> >
> >
> > [5] Sohn K, Berthelot D, Carlini N, et al. Fixmatch: Simplifying semi-supervised learning with consistency and confidence. NeurIPS, 2020.
> >
> > [6] Zhang, Bowen and Wang, Yidong and Hou, Wenxin and Wu, Hao and Wang, Jindong and Okumura, Manabu and Shinozaki, Takahiro. Flexmatch: Boosting semi-supervised learning with curriculum pseudo labeling. NeurIPS, 2021.
> >
> > [7] Mamshad Nayeem Rizve, Kevin Duarte, Yogesh S Rawat, Mubarak Shah. In Defense of Pseudo-Labeling: An Uncertainty-Aware Pseudo-label Selection Framework for Semi-Supervised Learning. ICLR 2021.
> >
> > [8] Bengio, Yoshua, Jérôme Louradour, Ronan Collobert, and Jason Weston. "Curriculum learning." In Proceedings of the 26th annual international conference on machine learning, pp. 41-48. 2009.
> >
> > [9] Zhou, Tianyi, Shengjie Wang, and Jeff Bilmes. "Robust curriculum learning: From clean label detection to noisy label self-correction." In International Conference on Learning Representations. 2020.

---

> ### Author Response · Authors · 2023-11-19
> **Response to Reviewer H8zS**
>
> ## REUTERS-10K dataset
>
> As shown in Table 1 of [1], the REUTERS-10k dataset contains only 10,000 points and 4 classes, which is significantly smaller than the ImageNet-R and iNature500 datasets used in our experiments. Additionally, it is a dataset intended for document clustering, whereas our manuscript focuses on image clustering.
>
> [1] Xie, Junyuan, Ross Girshick, and Ali Farhadi. "Unsupervised deep embedding for clustering analysis." International conference on machine learning. PMLR, 2016
>
> ## Why it can be called an unbalanced OT problem?
>
> When $F_1$ and $F_2$ are equality constraints, Eq. 1 is typically referred to as "classical" optimal transport. To generalize "classical" optimal transport, unbalanced OT [1,2,3] defines the problem on arbitrary positive measures and some divergence functions. In our manuscript, we refer to Eq. 3 as an unbalanced OT in order to distinguish it from "classical" optimal transport.
>
>
> [1] Lenaic Chizat, Gabriel Peyré, Bernhard Schmitzer, and François-Xavier Vialard. Scaling algorithms for unbalanced optimal transport problems. Mathematics of Computation, 87(314):2563–2609, 2018.
>
> [2] Matthias Liero, Alexander Mielke, and Giuseppe Savar ́e. Optimal entropy transport problems and a new Hellinger–Kantorovich distance between positive measures. Inventiones Mathematicae, pages 1–149, 2015.
>
> [3] Séjourné, Thibault, Gabriel Peyré, and François-Xavier Vialard. "Unbalanced Optimal Transport, from theory to numerics." Handbook of Numerical Analysis 24 (2023): 407-471.
>
>
> ## Mini-Batch
>
> Please refer to General Response 3.
>
>
>
> ## NMI explanation
>
> Our method does achieve better embedding but due to the calculation of NMI, we achieve modest improvement.
>
> Baselines tend to merge tail classes into head classes, whereas our approach tends to divide head classes into multiple clusters, resulting in a more even class distribution (Figure S7). The formula for the Normalized Mutual Information (NMI) is $\frac{2I(Y;C)}{H(Y)+H(C)}$, where Y represents the ground truth and C denotes the cluster label. Despite our method achieving superior clustering performance, as indicated by a higher $I(Y;C)$, it also results in a relatively uniform distribution, thereby increasing $H(C)$. This leads to only a modest improvement or decrease in the NMI metric.
>
>
>
>
> ## Explanation of Figure 4
> In Figure 4, we show the weighted precision and recall based on our selection strategy, which generates a weight, $Q1_K$, for samples. As training progresses, we increase $\rho$ using a sigmoid ramp-up strategy and select more samples, which inevitably includes more noisy samples, leading to a decrease in weighted precision and recall after 10 epochs. However, the precision and recall of the entire dataset are steadily increasing, illustrating our method learns meaningful clusters gradually. We have analyzed this phenomenon in the ablation study and mentioned that the current ramp-up function may not be optimal, leaving more advanced designs for future work.

---

> > ### Comment · Reviewer_H8zS · 2023-11-23
> > **Response to the Rebuttal**
> >
> > Thanks for the responses. My concerns have been addressed. So I increase my score.

---

> ### Author Response · Authors · 2023-11-23
> **Response to Reviewer H8zS**
>
> Thank you so much for taking the time to review our submission. We appreciate your detailed feedback and appraisal of our work, which we have taken into careful consideration in our rebuttal response. As the rebuttal process is coming to an end, we would be grateful if you could acknowledge receipt of our responses and let us know if they address your concerns. We remain eager to engage in any further discussions.

---

### Author Response · Authors · 2023-11-19
**General Response Part 1**

We thank the reviewers' valuable feedback and are encouraged by "the proposed algorithm is overall reasonable" (Reviewer H8zS), "this paper studies an interesting problem" (Reviewer xbkw), and "the technical point is good" (Reviewer qtAk). In the following, we address the common concerns in this part and individual concerns separately.

## Comparison and analysis on balanced training dataset
We emphasize that we tackle challenging and realistic imbalanced scenarios. In balanced settings, we believe that it is a bit unfair to compare our proposed method to methods developed in balanced scenarios, which usually take advantage of the uniform prior.

To compare and analyze our method with baselines on balanced datasets, we conduct experiments using the balanced CIFAR100 dataset. Overall, we have achieved superior performance on imbalanced datasets, and on balanced datasets, we have achieved satisfactory results.

|  Method   | Backbone | Fine-Tune | Confidence Ratio | Time Cost (h) | CIFAR100 |      |      |
| :-------: | :------: | :-------: | :--------------: | :-----------: | :------: | :--: | :--: |
|           |          |           |                  |               |   ACC    | NMI  | ARI  |
| SPICE$_s$ | ViT-B16  |   False   |        -         |      3.6      |   31.9   | 57.7 | 22.5 |
| SPICE$_s$ | ViT-B16  |   True    |        -         |      6.7      |   60.9   | 69.2 | 46.3 |
|   SPICE   | ViT-B16  |   False   |        >0        |      3.6      |   Fail   | Fail | Fail |
|   SPICE   | ViT-B16  |   True    |       >0.6       |      6.7      |   Fail   | Fail | Fail |
|   SPICE   | ViT-B16  |   True    |       0.5        |     21.6      |   68.0   | 77.7 | 54.8 |
|   Ours    | ViT-B16  |   False   |        -         |      2.9      |   61.9   | 72.1 | 48.8 |

*Note that, as imagenet-r and iNaturelist2018 subsets are inherently imbalanced, we do not conduct experiments on them.*

For all methods, we employ the ImageNet-1K pre-trained ViT-B16 backbone. Fine-tune refers to the first training stage of SPICE, which trains the model on the target dataset using self-supervised learning, like MoCo. SPICE$_s$ corresponds to the second stage of training, which fixes the backbone and learns the clustering head by selecting top-K samples for each class, heavily relying on the uniform distribution assumption. SPICE refers to the final joint training stage, which selects an equal number of reliable samples for each class based on confidence ratio and applies FixMatch-like training. In SPICE, the confidence ratio varies across different datasets. Time cost denotes the accumulated time cost in hours on A100 GPUs.

The observed results in the table reveal several key findings:
1. Without self-supervised training on the target dataset, SPICE$_s$ yields inferior results.
2. SPICE tends to struggle without self-supervised training and demonstrates sensitivity to hyperparameters.
3. SPICE incurs a prolonged training time.
4. Our method achieves comparable results with SPICE$_s$ and inferior results than SPICE.

On imbalanced datasets, as evidenced in Table S4 in Appendix F, SPICE encounters difficulties in clustering tail classes due to the noisy top-K selection, resulting in a lack of reliable samples for these classes. In contrast, our method attains outstanding performance on imbalanced datasets without the need for further hyperparameter tuning.

In summary, the strength of SPICE lies in its ability to leverage the balanced prior and achieve superior results in balanced scenarios. However, SPICE exhibits weaknesses:
1. It necessitates multi-stage training, taking 21.6 hours and 7.45 times our training cost
2. It requires meticulous tuning of multiple hyperparameters for different scenarios and may fail without suitable hyperparameters, while our method circumvents the need for tedious hyperparameter tuning
3. It achieves unsatisfactory results on imbalanced datasets due to its strong balanced assumption.

In contrast, the advantage of our method lies in its robustness to dataset distribution, delivering satisfactory results on balanced datasets and superior performance on imbalanced datasets.

---

> ### Author Response · Authors · 2023-11-19
> **General Response Part 2**
>
> ## Different representation learning baselines
> We appreciate the reviewer's suggestion and conduct additional experiments with the representation learning baselines, including K-means clustering on DINO's representation and [2].
>
> Note that we exclude reporting results from [1] and [3] for the following reasons:
> 1. [1] prunes the ResNet, making it challenging to transfer to ViT
> 2. [3] has not yet open-sourced the code for kernel density estimation, which is crucial for its success.
>
> For a fair comparison, we adopt the pretrained ViT-B16 and finetune the last block of the ViT-B16 with the latest BCL [2]. The results of BCL and the original DINO are presented below.
>
>
> | Method | Backbone | CIFAR100 |      |      | ImageNet-R |      |      | iNature100 |      |      |    |
> | :----: | :------: | :------: | :--: | :--: | :--------: | :--: | :--: | :--------: | :--: | :--: | -- |
> |        |          |          | ACC  | NMI  |     F1     | ACC  | NMI  |     F1     | ACC  | NMI  | F1 |
> |  BCL   | ResNet18 |   20.2   | 40.7 | 15.2 |     -      |  -   |  -   |     -      |  -   |  -   |    |
> |  DINO  | ViT-B16  |   36.6   | 68.9 | 31.0 |    20.5    | 39.6 | 22.2 |    40.1    | 67.8 | 34.2 |    |
> |  BCL   | ViT-B16  |   35.7   | 66.0 | 29.9 |    20.7    | 40.0 | 22.4 |    41.9    | 67.2 | 35.4 |    |
> |  Ours  | ViT-B16  |   38.2   | 69.6 | 32.0 |    25.9    | 45.7 | 27.3 |    44.2    | 67.0 | 36.9 |    |
>
> For both BCL and DINO, which focus on representation learning, we apply K-means to cluster the imbalanced training set in the representation space. The results indicate the following:
> 1. DINO achieves satisfactory results, particularly on CIFAR100, where the dataset is akin to the ImageNet-1K used for pre-training
> 2. BCL exhibits marginal improvement or even a decrease compared to DINO, as DINO serves as a robust unsupervised pre-trained model
> 3. Our method significantly outperforms baseline and unsupervised pre-training models across a wide range of metrics and datasets.
>
> These results demonstrate the effectiveness of our approach in handling imbalanced data.
>
> We have appended the results to Appendix J and updated the self-supervised long-tailed learning into related work.
>
>
> [1] Jiang, Z., Chen, T., Mortazavi, B.J. and Wang, Z., 2021, July. Self-damaging contrastive learning. In International Conference on Machine Learning (pp. 4927-4939). PMLR.
>
> [2] Zhou, Z., Yao, J., Wang, Y.F., Han, B. and Zhang, Y., 2022, June. Contrastive learning with boosted memorization. In International Conference on Machine Learning (pp. 27367-27377). PMLR.
>
> [3] Liu, H., HaoChen, J.Z., Gaidon, A. and Ma, T., 2021. Self-supervised learning is more robust to dataset imbalance. In International Conference on Learning Representation, 2022.
>
> [4] Mathilde Caron, Hugo Touvron, Ishan Misra, Hervé Jégou, Julien Mairal, Piotr Bojanowski, and
> Armand Joulin. Emerging properties in self-supervised vision transformers. In Proceedings of the
> IEEE/CVF international conference on computer vision, pp. 9650–9660, 2021.
>
>
>
> ## Implementation of Mini-Batch OT
>
> To address the limitation of the mini-batch, we introduce a memory buffer that stores a substantial number of sample predictions (e.g., 5120) to ensure the existence of minority clusters. In particular, before inputting data into P$^2$OT in each iteration, we concatenate predictions from the memory buffer with the current batch predictions to enhance stability while preserving efficiency. Additional details can be found in the last sentence of Section 4.3 and Appendix F.

---

### Meta-Review · Area_Chair_SQzv · 2023-12-06

**Metareview:**

This paper presents a interesting algorithmic contribution, offering an innovative approach despite its relatively extensive numerical cost. The authors have conducted a broad array of convincing numerical experiments which showcase the credibility and applicability of their work. Notably, the rebuttal provided by the authors stands out for its strength, both in terms of clarifying the initial claims and in introducing new numerical evidence. This comprehensive response effectively addresses most of the issues raised during the review process. Given the depth of the experimental validation and the thoroughness of the authors in responding to feedback, I recommend the acceptance of this paper.

**Justification For Why Not Higher Score:**

This paper is rather between accept and reject imho.

**Justification For Why Not Lower Score:**

I found the rebuttal to be very convincing, which is the reason why I recommend acceptance.

---

### Decision · Program_Chairs · 2024-01-16

Accept (poster)